# STVG-R1: Incentivizing Instance-Level Reasoning and Grounding in Videos via Reinforcement Learning

**Xiaowen Zhang**[1,2], **Zhi Gao**[2,3], **Licheng Jiao**[1✉], **Lingling Li**[1], **Qing Li**[2✉],

[1]Xidian University [2]State Key Laboratory of General Artificial Intelligence, BIGAI,
[3]Beijing Institute of Technology
https://stvg-r1.github.io/

## Abstract

In vision–language models (VLMs), misalignment between textual descriptions and visual coordinates often induces hallucinations. This issue becomes particularly severe in dense prediction tasks such as spatial–temporal video grounding (STVG). Prior approaches typically focus on enhancing visual–textual alignment or attaching auxiliary decoders. However, these strategies inevitably introduce additional trainable modules, leading to significant annotation costs and computational overhead. In this work, we propose a novel visual prompting paradigm that avoids the difficult problem of aligning coordinates across modalities. Specifically, we reformulate per-frame coordinate prediction as a compact instance-level identification problem by assigning each object a unique, temporally consistent ID. These IDs are embedded into the video as visual prompts, providing explicit and interpretable inputs to the VLMs. Furthermore, we introduce STVG-R1, the first reinforcement learning framework for STVG, which employs a task-driven reward to jointly optimize temporal accuracy, spatial consistency, and structural format regularization. Extensive experiments on six benchmarks demonstrate the effectiveness of our approach. STVG-R1 surpasses the baseline Qwen2.5-VL-7B by a remarkable margin of 20.9% on m_IoU on the HCSTVG-v2 benchmark, establishing a new state of the art (SOTA). Surprisingly, STVG-R1 also exhibits strong zero-shot generalization to multi-object referring video object segmentation task, achieving a SOTA 47.3% $\mathcal{J}\&\mathcal{F}$ on MeViS.

## 1 Introduction

In video grounding task, hallucination in vision–language models (VLMs) is a common phenomenon, where timestamps may extend beyond video duration or coordinates may exceed the frame resolution (Wang et al., 2024a; Liu et al., 2024a; Chen et al., 2024). A widely accepted perspective is that the hallucinations stem from misalignments between the visual and textual modalities (Lin et al., 2024; Wang et al., 2024b). Such misalignment leads to greater performance degradation in dense prediction tasks, where bounding box or segmentation mask for each frame are required.

To reduce the impact of cross-modal misalignment, existing research focuses on improving the alignment capability of VLMs (Wang et al., 2025a; Ye et al., 2024) or avoiding direct coordinate prediction (Yuan et al., 2025; Sun et al., 2025). Despite their success, these strategies typically introduce additional learnable components and exhibit limited generalization. As illustrated in Figure 1, alignment-based approaches (Li et al., 2025) directly output explicit frame-level coordinates, but struggle in multi-object scenes and often yield inconsistent or even meaningless predictions, such as $[0.00, 0.00, 0.27, 0.00]$. In contrast, decoder–based methods alleviate this by introducing segmentation tokens for cross-frame consistent prediction, yet their implicit outputs limit generalization. Motivated by these challenges and prior attempts, we propose a core idea: if the complex per-frame coordinate prediction can be reformulated into a compact and interpretable formulation, it becomes possible to mitigate visual–textual misalignment and enhance generalization.

Based on the above observation, we draw inspiration from existing research on visual prompts, which demonstrates the effectiveness of simple yet consistent referential cues for object represen-

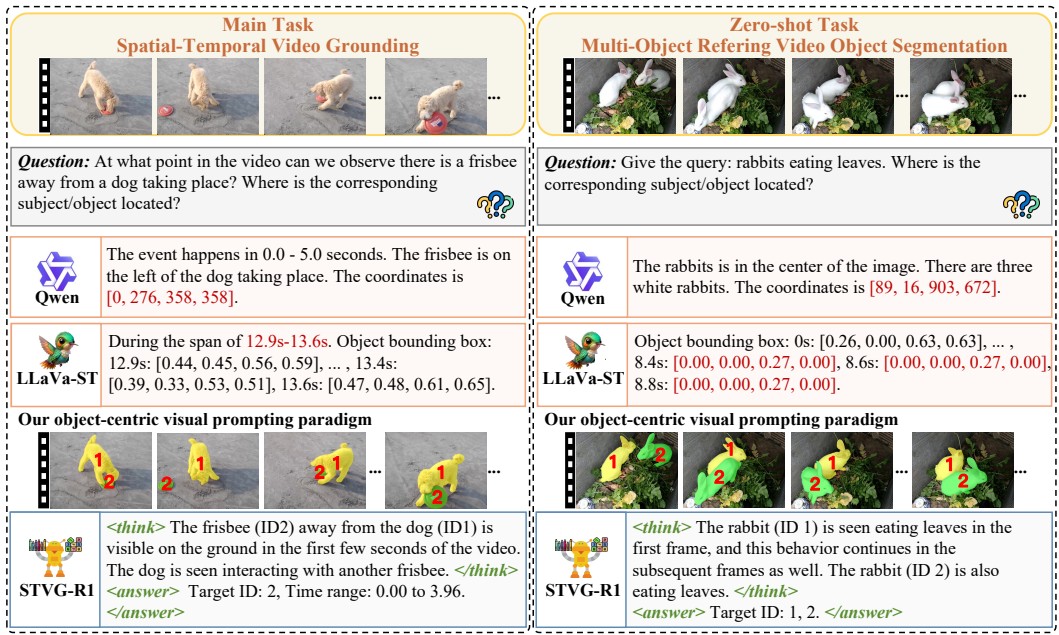

Figure 1: Comparisons of general VLMs, specialized VLMs, and proposed STVG-R1 model. While Qwen2.5-VL-7B outputs a single meaningless bounding box without timestamps, LLaVA-ST is restricted to one bounding box per frame. In contrast, STVG-R1 achieves strong performance on both spatial–temporal video grounding and zero-shot multi-object referring video object segmentation.

tation (Shtedritski et al., 2023; Cai et al., 2024; Yang et al., 2024). Taking GPT4Scene (Qi et al., 2025) as an example, consistent object IDs across multi-view images is embedded to enhance 3D understanding. Following this insight, in this paper, we introduce an object-centric visual prompting paradigm for spatial–temporal video grounding (STVG). Specifically, each object is automatically assigned a unique and temporally consistent identifier throughout the video sequence. Concretely, the first frame is processed with an object detector (Tian et al., 2025; Liu et al., 2024b; Xiao et al., 2023) to obtain candidate bounding boxes, which are further refined using the segmentation and tracking capabilities of SAM2 (Ravi et al., 2024). To handle newly appearing or previously missed objects, re-detection is performed at fixed intervals, and ReID is employed to maintain temporal consistency. Finally, each candidate instance is overlaid with a numeric marker that serves as its object ID on its center, yielding a compact yet interpretable formulation for video spatial grounding.

Building on this paradigm, we introduce STVG-R1, the first reinforcement learning framework for STVG. Unlike conventional supervised fine-tuning (SFT), which relies on token-level loss, STVG-R1 incorporates a task-driven reward that jointly optimizes temporal accuracy, spatial consistency, and structural correctness. A positive spatial consistency reward is obtained when the predicted object ID is aligned with the ground truth and also falls within the localized temporal segment.

The object-centric visual prompting paradigm achieves substantial performance improvements across four general VLMs in zero-shot settings. Specifically, InternVL3-8B (Zhu et al., 2025), Qwen2.5-VL-7B, Qwen2.5-VL-72B (Bai et al., 2025), and Qwen3-VL-8B improve vIoU@0.3 by +3.6%, +12.5%, +6.0%, and +28.3% on HCSTVG-v1 (Tang et al., 2021). Beyond zero-shot scenarios, the enhanced reasoning capability introduced by reinforcement learning establishes new SOTA on five benchmarks. Remarkably, the STVG-R1 also achieves strong performance on the unseen multi-object referring video object segmentation task, highlighting its robust generalization ability. We attribute this generalization to the object-centric visual prompts, which provide explicit object identifiers during reinforcement learning, enabling instance-level reasoning and grounding.

The main contributions of this paper are as follows: (1) We introduce a simple yet effective object-centric visual prompting paradigm that reformulates dense per-frame coordinate prediction into a compact object ID identification task. (2) We propose STVG-R1, the first reinforcement learning framework for spatial–temporal video grounding, built upon the GRPO algorithm. (3) Extensive experiments across six benchmarks demonstrate the effectiveness of our approach. Moreover, its strong

Figure 2: Comparison of paradigms: (a) VLM produces both timestamps and frame-level coordinates with a **trainable** alignment block; (b) VLM generates segmentation tokens, which are then processed by a **trainable** decoder; (c) our method uses **training-free** object-centric visual prompted video for spatial-temporal video grounding.

performance on the unseen multi-object referring video object segmentation task further highlights its generalization capability.

## 2 RELATED WORK

### 2.1 SPATIAL TEMPORAL VIDEO GROUNDING

In the research on spatial–temporal video grounding, existing approaches can be broadly categorized into models based on visual–language pretraining (VLP) and models leveraging VLMs. VLP-based methods typically employ pretrained encoders, such as CLIP (Radford et al., 2021), I3D (Carreira & Zisserman, 2017), InternVideo-v2 (Wang et al., 2024c), and BERT (Devlin et al., 2019), to extract visual and textual features, followed by the design of task-specific modules for multimodal feature fusion and tailored decoding. These approaches (Gu et al., 2024; 2025; Lin et al., 2022) still demonstrate dominant performance on several STVG benchmarks (Tang et al., 2021; Zhang et al., 2020). However, despite their effectiveness, these VLP-based task-specific models continue to struggle with generalization, even on simpler spatial-only or temporal-only video grounding tasks.

Recent efforts have increasingly adopted VLMs (Li et al., 2024; Bai et al., 2025; Abdin et al., 2024; Zhang et al., 2024) for video spatial grounding, owing to their superior cross-modal reasoning and generalization abilities. Within this line of research, as shown in Figure 2(a), one direction directly exploits VLMs for dense prediction, producing both temporal segments and frame-level spatial localization results. For example, LLaVA-ST (Li et al., 2025) enhances the alignment between textual descriptions and visual coordinates by incorporating additional tokens into the input text embeddings. Subsequently, SpaceVLLM (Wang et al., 2025a) follows a similar strategy by introducing spatial-temporal query tokens to address the alignment challenge. However, these additional trainable tokens require large-scale, high-quality dense prediction data and lead to substantial computational overhead. As shown in Figure 2(b), another direction mitigates the impact of misalignment by prompting VLMs to generate segmentation tokens (Yuan et al., 2025; Sun et al., 2025; Munasinghe et al., 2025), which are then passed into a trainable decoder (Ravi et al., 2024). However, their reliance on iterative decoding further increases training complexity and time.

### 2.2 REINFORCEMENT LEARNING IN VLMS

Reinforcement learning (RL) has demonstrated strong potential in improving the reasoning capabilities of LLMs, particularly through reinforcement learning with verifiable reward (RLVR) (Guo et al., 2025; Chen et al., 2025a; Jaech et al., 2024). For VLMs, many works (Liu et al., 2025; Shen et al., 2025; Zhang et al., 2025a; Chen et al., 2025b; Zhang et al., 2025b) also apply this reward-driven training paradigm to tackle complex tasks (Yang et al., 2025; Fu et al., 2025). Recent studies further extend this direction to video understanding and multimodal agents (Fan et al., 2024; 2025). More closely related to our setting, Video-R1 (Feng et al., 2025) is the first attempt to explore the R1 paradigm in the video domain, introducing the T-GRPO algorithm to explicitly encourage temporal understanding by shuffling the order of input video frames. Building on this foundation, Time-R1 (Wang et al., 2025b) proposes a novel post-training framework for temporal video grounding, also based on the Group Relative Policy Optimization (GRPO) algorithm (Shao et al., 2024). More encouragingly, Time-R1 demonstrates that using continuous metrics such as IoU as rewards provides more intuitive optimization signals and achieves better performance than token-level supervised fine-tuning. However, applying RL to jointly address spatial–temporal video grounding

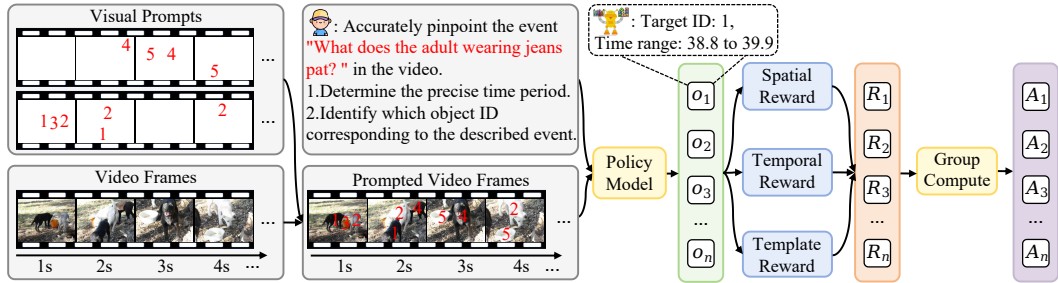

Figure 3: An illustration of our proposed STVG-R1 framework. Each object is assigned a unique ID via visual prompts, and the policy model is trained with spatial, temporal, and template rewards.

remains an underexplored yet promising direction.To bridge this gap, STVG-R1 integrates GRPO with STVG-specific rewards, achieving superior performance.

## 3 METHOD

### 3.1 STVG-R1 FRAMEWORK

Our approach reformulates spatial–temporal video grounding as a paradigm shift from dense per-frame bounding box regression to a compact formulation based on visual prompts. Figure 3 illustrates the overall architecture of the STVG-R1 model. Specifically, given a video $\mathcal{V} = \{I_1, \ldots, I_T\}$ with $T$ frames, each frame $I_t$ is first augmented with a set of visual prompts,

$$\mathcal{P}_t = \{p_1^t, \ldots, p_{K_t}^t\}, \qquad \tilde{I}_t \triangleq I_t \oplus \mathcal{P}_t, \tag{1}$$

where $t \in \{1, \ldots, T\}$ indexes frames, $K_t$ is the number of instances in frame $t$, and $\oplus$ denotes overlaying the visual prompts onto the frame $I_t$, yielding the augmented sequence $\tilde{\mathcal{V}} = \{\tilde{I}_1, \ldots, \tilde{I}_T\}$.

To control memory consumption, we constrain each video to approximately $R = 1.6 \times 10^6$ pixels in total. Concretely, for a video $\mathcal{V}$ with frame resolution $H \times W$ and duration $D$ seconds, we resize frames to $H' \times W' \approx R/(2D)$, where frames are uniformly sampled at 2 FPS. For example, a 30-second video yields 60 frames, each with a resolution of about $96 \times 96 \times 3$. Finally, the sequence of visual prompt–augmented frames $\tilde{\mathcal{V}}$ and a textual query $q$ are fed into a VLM $\pi_\theta$, which jointly predicts the temporal interval $[t_s, t_e]$ and the corresponding object identifier $\imath$.

### 3.2 OBJECT-CENTRIC PROMPTED VIDEO CONSTRUCTION

**Data format.** Each sample is denoted as $\{\mathcal{V}, q, P, M, A\}$, where $P = \{\mathcal{P}_t\}_{t=1}^T$ represents the set of visual prompts over frames, $M$ is the segmentation mask database, and $A$ is the ground-truth answer, defined as the target object ID. Concretely, $M$ stores for each frame $I_t$ a set of instance IDs paired with their run-length encoded masks (Golomb, 1966). For consistency with the ground-truth annotations, each mask $m_k^t$ is further converted into its corresponding bounding box $b_k^t$.

To formally derive $A$, we establish a frame-level correspondence within the ground truth temporal interval. For each frame $I_t$, we compute the IoU between the ground-truth $g_t$ and all candidate bounding boxes $\{b_k^t\}_{k=1}^{K_t}$, and assign to frame $t$ the ID $\imath_t$ with the highest overlap:

$$\imath_t = \arg \max_{k \in \{1, \ldots, K_t\}} \text{IoU}(g_t, b_k^t). \tag{2}$$

Over the entire video $\mathcal{V}$, the final answer $A$ is obtained by majority voting:

$$A = \arg \max_i \sum_{t=1}^T \mathbf{1}[\imath_t = i], \tag{3}$$

where $\mathbf{1}[\cdot]$ denotes the indicator function. This defines the target object as the identifier with the highest overall IoU consistency across the video.

**Data generation pipeline.** To construct object-centric visual prompted videos, we integrate several existing vision models into a unified pipeline. The first frame $I_1$ of each video is processed by an off-the-shelf object detector to produce bounding boxes $\{b_k^1\}_{k=1}^{K_1}$ for all candidate instances across COCO categories. These detections serve as prompts for SAM2, which generates high-quality segmentation masks $\{m_k^1\}_{k=1}^{K_1}$ that are then propagated to subsequent frames via tracking. To capture newly appearing or previously missed objects, periodic re-detection with IoU-based matching is performed, where each detection is compared against the masks already tracked to the same frame and is treated as a new instance only when its geometric overlap with all existing objects remains consistently low. Once a new instance is identified, SAM2 is further applied to perform both forward and backward tracking from the discovery frame to recover its complete temporal trajectory. For each mask $m_k^t$, we embed a compact visual prompt $p_k^t$ at its centroid $(x_k^t, y_k^t)$.

Importantly, although the COCO taxonomy does not fully cover all categories present in the videos, the detector can still provide bounding boxes for nearly all instances. For example, while fish is absent from the COCO label set, such instances are nevertheless detected under alternative categories. These semantic misclassifications do not affect our framework, as supervision depends only on consistent instance identities rather than precise class labels.

**Data source.** Two widely adopted STVG datasets are used for training. HCSTVG (Tang et al., 2021) focuses on human-centric grounding data. We merge the training splits of v1 and v2, and remove any samples that appear in the validation or test sets. VidSTG (Zhang et al., 2020) covers both humans and objects with diverse query types, providing both visual and linguistic diversity.

**Preprocessing Pipeline Robustness Analysis.** The reliability of the object-centric visual prompting pipeline is critical for stable reinforcement learning. We analyze potential failure modes in the object-centric prompting pipeline that could disrupt consistent instance identity construction. Global detection failures, defined as cases where the target object is not detected in any frame of the video, occur in fewer than 1% of all samples, indicating that the majority of target objects can be detected and assigned instance IDs. To address local missing detections caused by occlusion or fast motion, the pipeline incorporates periodic re-detection and SAM2's bidirectional propagation to recover full object trajectories. Identity consistency is further reinforced via majority voting during ID assignment, while a lightweight ID-repair step at evaluation time resolves occasional re-identification inconsistencies, as detailed in Appendix A.5. Together, these mechanisms mitigate detection and tracking noise, yielding stable instance identities for downstream optimization.

## 3.3 ENHANCING VLMs WITH REINFORCEMENT LEARNING

Since dense per-frame prediction is reformulated as a compact instance-level identification task, reinforcement learning can be directly applied to optimize the policy model with task-specific rewards. These identifiers further enable the model to produce more precise and interpretable reasoning chains during RL training, leading to more coherent spatial–temporal predictions.

**Reward modeling.** Building on DeepSeek-R1 (Guo et al., 2025), the reward design in STVG-R1 integrates both accuracy and format components. The accuracy reward measures the correctness of predictions, while the format reward enforces structural compliance with a predefined reasoning template. To capture both temporal and spatial accuracy, the accuracy reward is further decomposed into a temporal IoU reward and a spatial consistency reward. The temporal IoU reward quantifies the overlap between the predicted interval $[t_s, t_e]$ and the ground-truth segment $[t_s', t_e']$, defined as:

$$r_t(o) = \frac{[t_s, t_e] \cap [t_s', t_e']}{[t_s, t_e] \cup [t_s', t_e']}, \tag{4}$$

where $A \cap B$ and $A \cup B$ denote the intersection and union of intervals $A$ and $B$, respectively.

The spatial consistency reward verifies whether the predicted object ID is correct and appears within the localized temporal segment:

$$r_s(o) = \begin{cases} 1, & \text{if } \imath = \imath^* \text{ and } \imath \text{ appears in } [t_s, t_e], \\ 0, & \text{otherwise,} \end{cases} \tag{5}$$

where $\imath$ and $\imath^*$ denote the predicted and ground-truth object ID. This design is consistent with the vIoU metric in STVG, defined as $|P_u|^{-1}\sum_{t \in P_i} \text{IoU}(b_t, b_t^*)$, where $P_i$ and $P_u$ are the intersection

and union of the predicted and ground-truth temporal segments, and $b_t$ and $b_t^*$ are the predicted and ground-truth bounding boxes at frame $t$. Since vIoU jointly evaluates temporal and spatial accuracy, constraining the predicted ID to fall within the localized segment prevents trivial solutions and mitigates overfitting to dataset-specific temporal patterns, improving optimization stability.

Beyond accuracy, the format reward $r_f(o)$ enforces compliance with the predefined reasoning structure, encouraging the model to explicitly generate its reasoning process. A value of 1 is assigned only if the response encloses the reasoning within `<think>...</think>` and the final prediction within `<answer>...</answer>`. Reasoning traces with timestamps and instance IDs provide clearer references and more precise grounding.

The overall reward is the sum of the three components:

$$R(o) = r_t(o) + r_s(o) + r_f(o). \tag{6}$$

### 3.4 TRAINING STRATEGIES

The training objective is to optimize the policy model $\pi_\theta$ with GRPO (Guo et al., 2025), which has shown strong effectiveness in tasks with well-defined evaluation signals. Given a video–text query pair $(\tilde{\mathcal{V}}, q)$ sampled from the training distribution $\mathcal{D}$, the model generates $n$ candidate responses $o = \{o_1, \dots, o_n\}$, each assigned a reward $R(o_i)$ as defined in Equation 6. In our experiments, $n$ is set to 8. To normalize rewards within a group, the advantage $A_i$ of each response is computed as:

$$A_i = \frac{R(o_i) - \text{mean}(\{R(o_j)\}_{j=1}^n)}{\text{std}(\{R(o_j)\}_{j=1}^n)}. \tag{7}$$

The policy update objective encourages the current policy $\pi_\theta$ to assign higher probabilities to responses with larger normalized advantages, relative to the previous policy $\pi_{\theta_{\text{old}}}$. Formally, the optimization objective is defined as:

$$\mathcal{J}_{\text{GRPO}}(\theta) = \mathbb{E}_{(\tilde{\mathcal{V}},q)\sim\mathcal{D}, \{o_i\}_{i=1}^n \sim \pi_{\theta_{\text{old}}}(\cdot|q)}$$
$$\left[ \frac{1}{n} \sum_{i=1}^n \left( \min\left( \frac{\pi_\theta(o_i|q)}{\pi_{\theta_{\text{old}}}(o_i|q)} A_i, \text{clip}\left( \frac{\pi_\theta(o_i|q)}{\pi_{\theta_{\text{old}}}(o_i|q)}, 1-\epsilon, 1+\epsilon \right) A_i \right) - \beta D_{\text{KL}}(\pi_\theta \| \pi_{\text{ref}}) \right) \right], \tag{8}$$

where $\epsilon$ is the clipping parameter, $\beta$ controls the strength of KL regularization, and $\pi_{\text{ref}}$ denotes the frozen reference policy. The clipping term prevents excessively large updates, while the KL penalty constrains policy drift, together stabilizing optimization.

## 4 EXPERIMENTS

### 4.1 SETTING

**Implementation details.** The object detector used during our training and evaluation is YOLOv12-x (Tian et al., 2025) with a confidence threshold of 0.25, and SAM2.1-large is the segmentation and tracking model. Re-detection is performed every 15 frames, where a detection is treated as a new instance only if its IoU with all tracked objects is below 0.4 and its overlap ratio with all tracked objects is below 0.6. We employ the Qwen2.5-VL-7B (Bai et al., 2025) as the pre-trained model. We use AdamW (Loshchilov & Hutter, 2017) optimizer with a linear learning rate scheduler. The learning rate is $1.0e-6$ and the batch size is 1 per device. The model is trained for 1 epoch on our object-centric visual prompting dataset, and all experiments are conducted on 8×A100 GPUs.

**Benchmarks.** For STVG, HCSTVG-v1 and HCSTVG-v2 (Tang et al., 2021) are widely used for human-centric grounding, while ST-Align (Li et al., 2025) extends evaluation to both humans and objects and supports spatial video grounding (SVG). To capture fine-grained spatial understanding, MeViS (Ding et al., 2023) evaluates mask-level grounding under complex multi-object scenarios. Beyond STVG and SVG, we also consider video temporal grounding (VTG) with Charades-STA (Gao et al., 2017) and TVGBench (Wang et al., 2025b) to evaluate generalization.

**Evaluation metrics.** For STVG, following (Yang et al., 2022; Gu et al., 2024), we report m_tIoU for temporal localization accuracy and m_vIoU, vIoU@R for joint spatial–temporal grounding quality. For VTG, we adopt m_tIoU and tIoU@R. For mask-level referring video object segmentation, $\mathcal{J}$ is used to assess region similarity and $\mathcal{F}$ to measure contour accuracy.

Table 1: Performance comparison with state-of-the-art models on HCSTVG-v1 *test* set and HCSTVG-v2 *val* set (%). The results of GroundingGPT-7B are reported from SpaceVLLM, while those of InternVL3-8B, Qwen2.5-VL-7B, Qwen2.5-VL-72B and Qwen3-VL-8B are generated by our experiments. The best and second-best results are shown in **bold** and underlined.

| Models | HCSTVG-v1 | | | | HCSTVG-v2 | | | |
|---|---|---|---|---|---|---|---|---|
| | m_tIoU | m_vIoU | vIoU@0.3 | vIoU@0.5 | m_tIoU | m_vIoU | vIoU@0.3 | vIoU@0.5 |
| TubeDETR | - | 32.4 | 49.8 | 23.5 | 53.9 | 36.4 | 58.8 | 30.6 |
| STVGFormer | - | 36.9 | 62.2 | 34.8 | 58.1 | 38.7 | 65.5 | 33.8 |
| CG-STVG | 52.8 | 38.4 | 61.5 | 36.3 | 60.0 | 39.5 | 64.5 | 36.3 |
| TA-STVG | 53.0 | 39.1 | 63.1 | 36.8 | 60.4 | 40.2 | 65.8 | 36.7 |
| GroundingGPT-7B | 22.2 | 16.7 | 15.0 | 4.9 | 19.6 | 14.7 | 16.6 | 3.1 |
| SpaceVLLM-7B | **56.9** | **39.3** | 66.6 | 36.9 | 58.0 | 34.0 | 56.9 | 24.7 |
| InternVL3-8B | 22.6 | 11.7 | 15.3 | 2.8 | 24.9 | 12.8 | 14.2 | 3.2 |
|   +*VisualPrompt* | 22.9 ↑+0.3 | 13.8 ↑+2.1 | 18.9 ↑+3.6 | 4.2 ↑+1.4 | 25.0 ↑+0.1 | 15.4 ↑+2.6 | 18.1 ↑+3.9 | 4.5 ↑+1.3 |
| Qwen2.5-VL-7B | 40.3 | 19.7 | 28.2 | 7.9 | 45.1 | 19.3 | 26.0 | 8.2 |
|   +*VisualPrompt* | 38.7 ↓-1.6 | 24.8 ↑+5.1 | 40.7 ↑+12.5 | 13.4 ↑+5.5 | 44.7 ↓-0.4 | 19.5 ↑+0.2 | 28.7 ↑+2.7 | 10.9 ↑+2.7 |
| Qwen2.5-VL-72B | 40.7 | 23.9 | 37.0 | 15.1 | 43.9 | 23.4 | 36.1 | 13.4 |
|   +*VisualPrompt* | 38.8 ↓-1.9 | 26.0 ↑+2.1 | 43.0 ↑+6.0 | 15.2 ↑+0.1 | 42.0 ↓-1.9 | 27.3 ↑+3.9 | 43.5 ↑+7.4 | 16.3 ↑+2.9 |
| Qwen3-VL-8B | 48.6 | 19.5 | 28.2 | 10.6 | 51.2 | 17.2 | 22.8 | 7.6 |
|   +*VisualPrompt* | 48.7 ↑+0.1 | 33.0 ↑+13.5 | 56.5 ↑+28.3 | 27.3 ↑+16.7 | 52.2 ↑+1.0 | 35.8 ↑+18.6 | 57.9 ↑+35.1 | 30.3 ↑+22.7 |
| STVG-R1 | **56.9** | 39.1 | **66.7** | **38.6** | **61.3** | **40.8** | **67.9** | **38.8** |

Table 2: Performance comparison with state-of-the-art models on ST-Align benchmark (%). The results of Qwen2.5-VL-7B are generated by our experiments.

| Models | Spatial-Temporal Video Grounding | | | | Video Spatial Grounding | | |
|---|---|---|---|---|---|---|---|
| | tIoU@0.5 | m_tIoU | vIoU@0.5 | m_vIoU | vIoU@0.3 | vIoU@0.5 | m_vIoU |
| GroundingGPT-7B | 7.1 | 12.2 | 2.9 | 9.2 | 19.7 | 5.4 | 17.9 |
| VTimeLLM-7B | 7.1 | 15.5 | - | - | - | - | - |
| Grounded-VideoLLM-7B | 30.0 | 33.0 | - | - | - | - | - |
| LLava-ST-7B | **44.6** | 43.8 | 21.1 | 22.8 | 47.2 | 30.9 | 32.5 |
| Qwen2.5-VL-7B | 35.2 | 37.4 | 17.1 | 14.3 | 44.6 | 39.5 | 35.5 |
|   +*VisualPrompt* | 36.0 ↑+0.8 | 38.3 ↑+0.9 | 21.5 ↑+4.4 | 19.5 ↑+5.2 | 57.7 ↑+13.1 | 51.4 ↑+11.9 | 46.6 ↑+11.1 |
| STVG-R1 | 43.6 | **45.1** | **25.9** | **23.4** | **60.3** | **53.9** | **48.6** |

## 4.2 EVALUATION RESULTS ON SPATIAL TEMPORAL VIDEO GROUNDING

Table 1 and Table 2 present results on HCSTVG-v1/v2 and ST-Align. TubeDETR (Yang et al., 2022), STVGFormer (Lin et al., 2023), CG-STVG (Gu et al., 2024), and TA-STVG (Gu et al., 2025) are four VLP-based specialized models. InternVL3-8B, Qwen2.5-VL-7B/72B, and Qwen3-VL-8B first perform temporal grounding to predict the frame range, and then apply spatial grounding on frames within the intersection of predicted $b_t$ and ground-truth $b_t^*$ for evaluation.

**Zero-shot.** The object-centric visual prompting paradigm outperforms the two-stage evaluation across InternVL3-8B, Qwen2.5-VL-7B, Qwen2.5-VL-72B and Qwen3-VL-8B, achieving m_vIoU scores of 15.4%, 19.5%, 27.3%, and 35.8% on HCSTVG-v2, respectively. The improvement can be attributed to the ability of our paradigm to leverage information from the entire video sequence when generating spatial predictions. However, temporal performance slightly declines for Qwen2.5-VL models due to occlusion of fine-grained details and distributional shifts introduced by visual prompts. Notably, Qwen3-VL-8B exhibits a significant performance boost under our visual prompting paradigm, as the baseline model often fails to detect any target when only static image inputs are provided for dynamically described scenes.

**Fine-tuning.** Reinforcement learning yields substantial gains in both temporal and spatial performance, establishing new state-of-the-art results on HCSTVG-v1, HCSTVG-v2, and ST-Align. On HCSTVG-v2, compared with the strongest SFT-trained VLM model SpaceVLLM, STVG-R1 achieves absolute improvements of 4.0%, 6.2%, 10.9% and 14.1% across four evaluation metrics. As shown in Table 2, STVG-R1 also surpasses the strongest ST-Align model LLaVA-ST by +0.6% on m_vIoU. These spatial gains highlight the effectiveness of our object-centric visual prompting paradigm in enforcing consistent object-level predictions, while reinforcement learning further enhances reasoning ability, leading to more coherent spatial–temporal video grounding.

## 4.3 EVALUATION RESULTS ON VIDEO SPATIAL GROUNDING

Since vIoU in STVG is inevitably affected by temporal prediction quality, we further evaluate video spatial grounding to isolate spatial capability. As shown in Table 2, the proposed object-centric visual prompting paradigm achieves a notable zero-shot gain of 11.1% on m_vIoU on ST-Align video spatial grounding. And after RL, STVG-R1 surpasses the second-best model LLaVA-ST by 13.1% on m_vIoU. More importantly, Table 3 reports the results on the multi-object referring video object segmentation task. STVG-R1 sets a new state-of-the-art of 47.3% on $\mathcal{J}\&\mathcal{F}$ on MeViS, de-

Table 3: Performance comparison with state-of-the-art models on MeViS (%). The results of TrackGPT are generated by VISA.

| Models | $J$ | $F$ | $J\&F$ |
|---|---|---|---|
| URVOS (Seo et al., 2020) | 25.7 | 29.9 | 27.8 |
| MTTR (Botach et al., 2022) | 28.8 | 31.2 | 30.0 |
| ReferFormer (Wu et al., 2022) | 29.8 | 32.2 | 31.0 |
| LMPM (Ding et al., 2023) | 34.2 | 40.2 | 37.2 |
| LISA (Lai et al., 2024) | 35.1 | 39.4 | 37.2 |
| TrackGPT (Stroh, 2024) | 37.6 | 42.6 | 40.1 |
| VISA (Yan et al., 2024) | 40.7 | 46.3 | 43.5 |
| VideoGlaMM (Munasinghe et al., 2025) | 42.1 | 48.2 | 45.2 |
| STVG-R1 | **44.7** | **50.0** | **47.3** |

spite being trained only on single-object STVG data. This demonstrates the strong generalization ability of our visual prompting paradigm, where the simplified identifier-based formulation facilitates transfer to more complex multi-object scenarios.

## 4.4 EVALUATION RESULTS ON VIDEO TEMPORAL GROUNDING

We further evaluate STVG-R1 on out-of-distribution video temporal grounding benchmarks. As shown in Table 4, STVG-R1 achieves the best zero-shot performance on Charades-STA, surpassing the second-best model LLaVA-ST by +7.7% at tIoU@0.5. Although slightly below the task-specific Time-R1, STVG-R1 achieves competitive results on TVGBench, highlighting strong generalization.

Table 4: Performance comparison with state-of-the-art models on Charades-STA and TVGBench (%). The results marked with ∗ represent models training on corresponding dataset, while others indicate zero-shot settings.

| Models | Charades-STA | | TVGBench | |
|---|---|---|---|---|
| | tIoU@0.3 | tIoU@0.5 | tIoU@0.3 | tIoU@0.5 |
| TimeSuite (Zeng et al., 2024) | 69.9 | 48.7 | 31.1 | 18.0 |
| TRACE (Guo et al., 2024) | - | 40.3 | 37.0 | 25.5 |
| LLaVA-ST (Li et al., 2025) | 63.1 | 44.8 | - | - |
| Time-R1 (Wang et al., 2025b) | 78.1∗ | 60.8∗ | 41.8 | **29.4** |
| STVG-R1 | **73.2** | **52.5** | **42.5** | 27.4 |

## 4.5 ABLATION

**Ablation on visual prompt design.** Following prior work (Cai et al., 2024; Shtedritski et al., 2023), red-colored visual prompts consistently yield superior performance in identifier recognition. As shown in Table 5, varying the **font size** leads to only moderate performance differences. Smaller prompts slightly improve temporal grounding, likely due to reduced visual occlusion. Overall, performance remains relatively stable across sizes, and a medium size provides a balanced trade-off between visibility and visual interference. Regarding **prompt types**, both letters and numbers

Table 5: Ablation study on visual prompt designs on HCSTVG-v1 with zero-shot Qwen2.5-VL-7B. U-Letters denotes uppercase letters, L-Letters denotes lowercase letters, and Mix refers to a combination of numbers and uppercase letters.

| Size | Type | m_tIoU | m_vIoU | vIoU@0.3 | vIoU@0.5 |
|---|---|---|---|---|---|
| 10 | Number | 38.1 | 24.6 | 39.4 | 12.0 |
| 20 | Number | 38.0 | **24.9** | **40.6** | 12.2 |
| 30 | Number | 37.5 | 24.1 | 38.9 | 11.8 |
| 40 | Number | 37.4 | 23.2 | 36.4 | 11.6 |
| 20 | U-Letters | **39.0** | 24.4 | 38.0 | **12.6** |
| 20 | L-Letters | 38.7 | 24.0 | 37.4 | 12.1 |
| 20 | Mix | 38.7 | 15.7 | 20.0 | 5.7 |

achieve comparable performance. Letters yield slightly stronger temporal grounding, possibly because they are single-character tokens, whereas multi-digit numbers occupy more space. In contrast, numbers provide marginally better spatial accuracy, likely due to their more consistent visual structure. A mixed design mapping larger numbers to letters does not provide additional benefits, suggesting that simple and consistent prompts are sufficient. Based on these observations, red-colored numeric prompts with font size 20 are adopted as the default configuration for all experiments.

**Ablation on mask filtering thresholds.** To reduce visual clutter from dense prompts, we apply mask filtering that removes small instances whose size falls below a fraction of the maximum mask within each category per frame. As shown in Table 6, higher thresholds $\theta$ degrade **data quality**, with m_vIoU dropping to 65.1% at 1/2 on HCSTVG-v1. In contrast, **zero-shot** evaluation with Qwen2.5-VL-7B shows that moderate filtering improves spatial grounding while maintaining temporal accu-

Table 6: Experimental results of mask filtering thresholds on HCSTVG-v1. Values before '/' denote the upper bound, and those after '/' are zero-shot results with Qwen2.5-VL-7B.

| $\theta$ | m_tIoU | m_vIoU | vIoU@0.3 | vIoU@0.5 |
|---|---|---|---|---|
| 0 | 100.0/37.5 | 69.1/23.3 | 97.2/37.4 | 89.9/11.9 |
| 1/4 | 100.0/38.0 | 68.3/24.6 | 95.8/40.5 | 88.5/11.6 |
| 1/3 | 100.0/38.0 | 67.7/24.9 | 94.6/40.6 | 87.6/12.1 |
| 1/2 | 100.0/38.1 | 65.1/24.1 | 89.9/38.2 | 83.5/11.4 |

racy. A threshold of 1/3 achieves the best trade-off between data quality and zero-shot performance, indicating that many small-scale objects are not semantically salient and may even introduce noise.

**Ablation on our modules.** We further conduct ablation studies to explore the individual contributions of object-centric visual prompting paradigm and reinforcement learning. Without visual prompts, GRPO follows the zero-shot two-stage evaluation and is optimized only with the temporal reward. As shown in Table 7 and Table 8, visual prompts primarily enhance spatial localization, while GRPO substantially improves temporal accuracy. VisualPrompt-SFT improves all metrics but slightly reduces temporal grounding on ST-Align, where reasoning is critical. The combination of visual prompt and GRPO yields the most consistent gains, achieving state-of-the-art performance.

Table 7: Ablation study with different modules on HCSTVG-v1 and HCSTVG-v2.

| Models | HCSTVG-v1 | | | | HCSTVG-v2 | | | |
|---|---|---|---|---|---|---|---|---|
| | m_tIoU | m_vIoU | vIoU@0.3 | vIoU@0.5 | m_tIoU | m_vIoU | vIoU@0.3 | vIoU@0.5 |
| Qwen2.5-VL-7B | 40.3 | 19.7 | 28.2 | 7.9 | 45.1 | 19.3 | 26.0 | 8.2 |
| +*VisualPrompt* | 38.7 | 24.8 | 40.7 | 13.4 | 44.7 | 19.2 | 28.7 | 10.9 |
| +*GRPO* | **57.5** | 24.7 | 37.8 | 17.6 | 61.4 | 24.2 | 37.7 | 15.4 |
| +*VisualPrompt-SFT* | 50.9 | 34.3 | 60.2 | 28.0 | 54.4 | 36.5 | 60.8 | 31.3 |
| +*VisualPrompt-GRPO* | 56.9 | **39.1** | **66.7** | **38.6** | **62.0** | **40.2** | **67.8** | **38.8** |

Table 8: Ablation study with different modules on ST-Align.

| Models | Spatial-Temporal Video Grounding | | | | Spatial Video Grounding | | |
|---|---|---|---|---|---|---|---|
| | tIoU@0.5 | m_tIoU | vIoU@0.5 | m_vIoU | vIoU@0.3 | vIoU@0.5 | m_vIoU |
| Qwen2.5-VL-7B | 35.2 | 37.4 | 17.1 | 14.3 | 44.6 | 39.5 | 35.5 |
| +*VisualPrompt* | 36.0 | 38.3 | 21.5 | 19.5 | 57.7 | 51.4 | 46.6 |
| +*GRPO* | 41.8 | 43.7 | 17.3 | 20.0 | - | - | - |
| +*VisualPrompt-SFT* | 34.6 | 36.6 | 21.5 | 19.3 | 58.6 | 52.3 | 47.2 |
| +*VisualPrompt-GRPO* | **43.6** | **45.1** | **25.9** | **23.4** | **60.3** | **53.9** | **48.6** |

**Ablation on reward design.** In STVG-R1, the temporal reward is directly derived from the evaluation metric, while the spatial reward adopts a sparse 0/1 formulation that provides credit only when the predicted instance ID matches the ground truth. To examine the role of the spatial component, we tested two alternatives: (1) a coupled spatio-temporal reward inspired by vIoU, defined as $R(o) = r_t(o) + r_s(o), r_t(o) + r_f(o)$, and (2) a continuous spatial reward computed as the average per-frame IoU over the temporal intersection, $r_s = \frac{1}{|\mathcal{T}_\cap|} \sum_{t \in \mathcal{T}_\cap} \text{IoU}(\hat{B}_t, B_t^*)$. Neither variant improves performance. On HCSTVG-v1, the coupled reward reduces m_vIoU from 39.1% to 38.3%, while the continuous spatial reward yields 38.6%. This suggests that the sparse spatial reward better aligns with the objective of selecting a single correct instance, avoiding additional noise.

## 4.6 IMPACT OF VISUAL PROMPT OCCLUSION

A potential concern is that overlaying numeric identifiers may introduce visual pollution, particularly for OCR-related tasks. To assess this effect, we evaluate Qwen2.5-VL-7B with and without visual prompts on the MME-VideoOCR Shi et al. (2025) benchmark, which covers ten OCR-centric tasks. As shown in Table 9, performance differences remain consistently small across all tasks. Notably, tasks that require fine-grained character recognition, such as TR, show a small performance drop from 69.6% to 69.3%. In contrast, higher-level tasks such as VTQA and TG improve from 76.4% to 77.4% and from 63.2% to 65.3%, suggesting that the added markers can help guide attention toward relevant regions. These results indicate that the visual prompts introduce negligible interference.

Table 9: Evaluation results on MME-VideoOCR. 'TR' denotes Text Recognition, 'VTQA' Visual Text QA, 'TG' Text Grounding, 'AR' Attribute Recognition, 'CDT' Change Detection & Tracking, 'STP' Special Text Parsing, 'CFTU' Cross-Frame Text Understanding, 'TBR' Text-Based Reasoning, 'TBVU' Text-Based Video Understanding, and 'RVT' Robust Video Testing.

| Method | TR | VTQA | TG | AR | CDT | STP | CFTU | TBR | TBVU | RVT | Total |
|---|---|---|---|---|---|---|---|---|---|---|---|
| Qwen2.5-VL-7B | **69.6** | 76.4 | 63.2 | **69.8** | 45.7 | **64.4** | **22.7** | 54.7 | **38.2** | **79.0** | **59.4** |
| + VisualPrompt | 69.3 | **77.4** | **65.3** | 68.5 | **46.2** | 64.1 | 22.7 | **54.8** | 38.2 | 78.8 | 58.9 |

## 4.7 VISUALIZATION

In Figure 4, we present a case of spatial–temporal video grounding with our object-centric prompting paradigm. The model first identifies the object IDs relevant to the query and then determines the temporal boundaries. During the reasoning process, complex descriptive appearance expressions are compactly mapped into object IDs, facilitating precise instance-level reasoning and grounding.

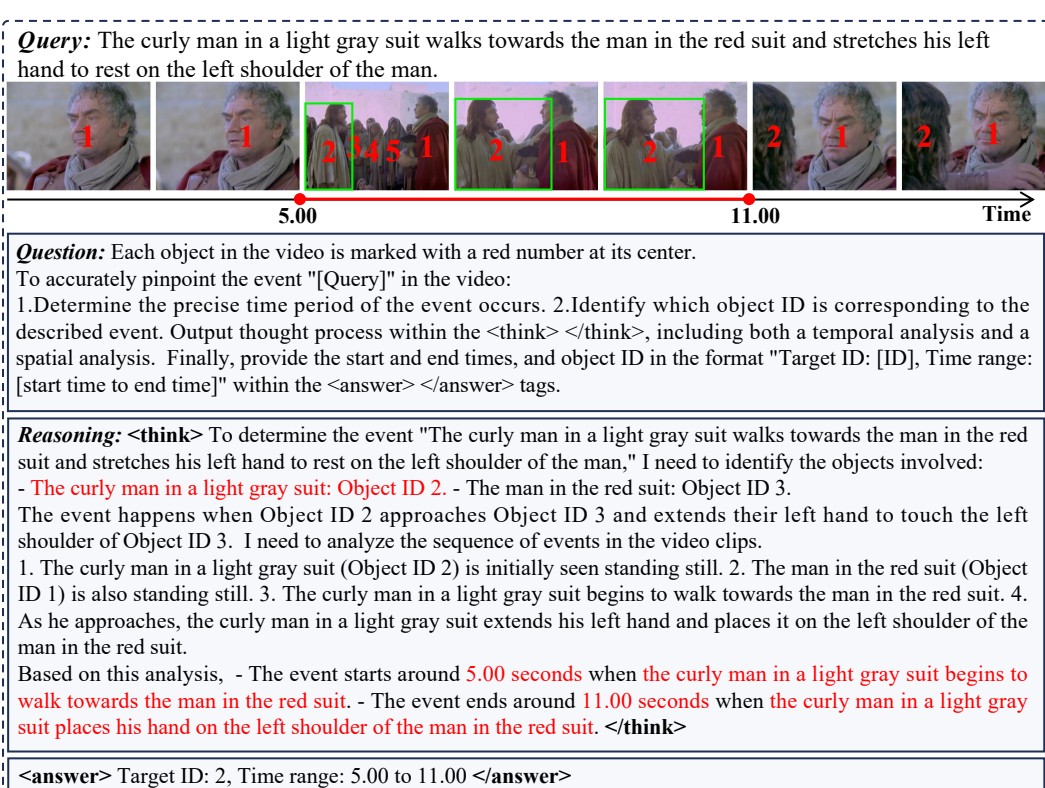

*Query:* The curly man in a light gray suit walks towards the man in the red suit and stretches his left hand to rest on the left shoulder of the man.

*Question:* Each object in the video is marked with a red number at its center.
To accurately pinpoint the event "[Query]" in the video:
1.Determine the precise time period of the event occurs. 2.Identify which object ID is corresponding to the described event. Output thought process within the <think> </think>, including both a temporal analysis and a spatial analysis. Finally, provide the start and end times, and object ID in the format "Target ID: [ID], Time range: [start time to end time]" within the <answer> </answer> tags.

*Reasoning:* <think> To determine the event "The curly man in a light gray suit walks towards the man in the red suit and stretches his left hand to rest on the left shoulder of the man," I need to identify the objects involved:
- The curly man in a light gray suit: Object ID 2. - The man in the red suit: Object ID 3.
The event happens when Object ID 2 approaches Object ID 3 and extends their left hand to touch the left shoulder of Object ID 3. I need to analyze the sequence of events in the video clips.
1. The curly man in a light gray suit (Object ID 2) is initially seen standing still. 2. The man in the red suit (Object ID 1) is also standing still. 3. The curly man in a light gray suit begins to walk towards the man in the red suit. 4. As he approaches, the curly man in a light gray suit extends his left hand and places it on the left shoulder of the man in the red suit.
Based on this analysis, - The event starts around 5.00 seconds when the curly man in a light gray suit begins to walk towards the man in the red suit. - The event ends around 11.00 seconds when the curly man in a light gray suit places his hand on the left shoulder of the man in the red suit. </think>

<answer> Target ID: 2, Time range: 5.00 to 11.00 </answer>

Figure 4: Case study of STVG-R1 on the spatial-temporal video grounding task.

## 5 CONCLUSION

This work addresses challenges of coordinates visual–textual misalignment and instance prediction inconsistency across videos. We propose an object-centric visual prompting paradigm that reformulates per-frame coordinate prediction into a compact and interpretable instance-level identification problem. We further introduce STVG-R1, a reinforcement learning framework optimized with task-driven rewards. Experiments across six benchmarks demonstrate the effectiveness of compact visual prompts and reinforcement learning in enhancing reasoning consistency and generalization. Future work will extend our detector-based framework from natural images to broader visual domains.

## 6 REPRODUCIBILITY STATEMENT

We make every effort to ensure the reproducibility of our work. Detailed descriptions of the model architecture, training pipeline, training datasets, and reward design for STVG-R1 are provided in Section 3. Implementation details are reported in Section 4.1. The design of visual prompts and filtering thresholds is described in Section 4.5, and the prompts used for training and evaluation across different tasks are presented in Section A.1.

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

# A APPENDIX

## A.1 PROMPT FOR TRAINING AND EVALUATION

The prompt for spatial-temporal video grounding is shown in Figure 5.

> Each object in the video is marked with a red number at its center.
>
> To accurately pinpoint the event "[Query]" in the video:
> 1.Determine the precise time period of the event occurs.
> 2.Identify which object ID is corresponding to the described event.
>
> Output thought process within the <think> </think>, including both temporal analysis and spatial analysis. Finally, provide the start and end times, and object ID in the format "Target ID: [ID], Time range: [start time to end time]" within the <answer> </answer> tags.

Figure 5: Prompt for spatial-temporal video grounding.

The prompt for video spatial grounding and referring video object segmentation is shown in Figure 6.

> Each object in the video is marked with a red number at its center.
>
> To accurately pinpoint the event "[Query]" in the video:
> 1.Identify which object ID is corresponding to the described event.
>
> Output thought process within the <think> </think>, including spatial analysis. Finally, provide the object ID in the format "Target ID: [ID]" within the <answer> </answer> tags.

Figure 6: Prompt for video spatial grounding and referring video object segmentation.

The prompt for video temporal grounding is shown in Figure 7.

> Each object in the video is marked with a red number at its center.
>
> To accurately pinpoint the event "[Query]" in the video:
> 1.Determine the precise time period of the event occurs.
>
> Output thought process within the <think> </think>, including temporal analysis. Finally, provide the start and end times in the format "Time range: [start time to end time]" within the <answer> </answer> tags.

Figure 7: Prompt for video temporal grounding.

## A.2 VISUALIZATIONS OF DIFFERENT TASKS

We provide more cases of spatial-temporal video grounding and referring video object segmentation tasks, as presented in Figure 8 and Figure 9. Although some visual prompts are filtered out in certain frames, the corresponding instances remain stored in the mask database, thereby enhancing the final spatial localization capability.

(a)

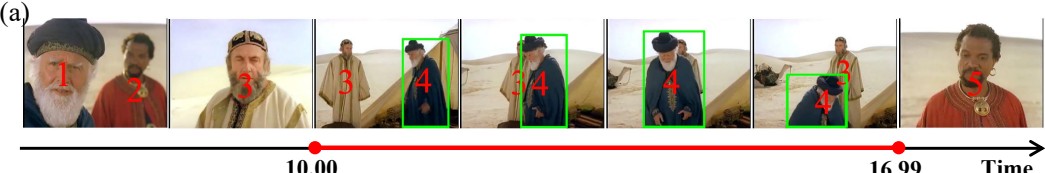

**10.00**                                                                **16.99**        **Time**

*Query:* The blue man walks a few steps forward as he speaks, and then sits down.

> *Reasoning:* <think> To determine the event "The blue man walks a few steps forward as he speaks, and then sits down," I need to identify the blue man first. The blue man is object ID 4. He wears a blue robe and a black hat. In the video, the blue man starts by standing still and then begins to walk forward while speaking. After walking a few steps, he stops and sits down on the ground. This sequence happens between 10.00 and 16.99 seconds. </think>
>
> <answer> Target ID: 4, Time range: 10.00 to 16.99 </answer>

(b)

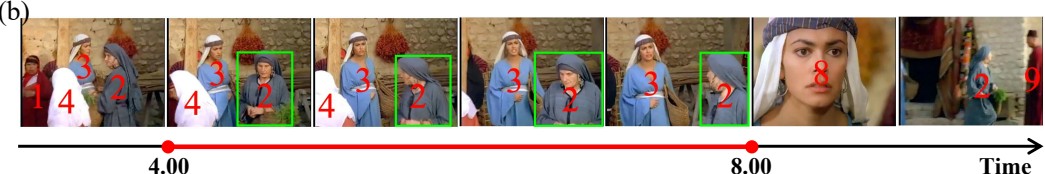

**4.00**                                                                **8.00**        **Time**

*Query:* The woman in the gray dress turns and takes a step forward, looking back at the woman wearing a white headscarf, and then takes another step back, looking at the woman wearing a white headscarf.

> *Reasoning:* <think> To determine the event "The woman in the gray dress turns and takes a step forward, looking back at the woman wearing a white headscarf, and then takes another step back, looking at the woman wearing a white headscarf," I need to analyze the sequence of actions:
>
> 1. The woman in the gray dress (ID 2) initially faces the woman in the white headscarf (ID 4).
> 2. She turns her body slightly towards the camera.
> 3. She takes a step forward while still facing the woman in the white headscarf.
> 4. She looks back at the woman in the white headscarf.
> 5. She takes another step back while still facing the woman in the white headscarf.
>
> Based on this sequence, the event occurs between the moments when she turns her body and takes the first step forward until she takes the second step back.
>
> Temporal Analysis:
> - The event starts around 4.00 seconds when the woman in the gray dress turns her body.
> - The event ends around 8.00 seconds when she takes the second step back.
>
> Spatial Analysis:
> - The woman in the gray dress (object ID 2) is the target object for the event. </think>
>
> <answer> Target ID: 2, Time range: 4.00 to 8.00 </answer>

Figure 8: Cases for spatial-temporal video grounding.

(a)

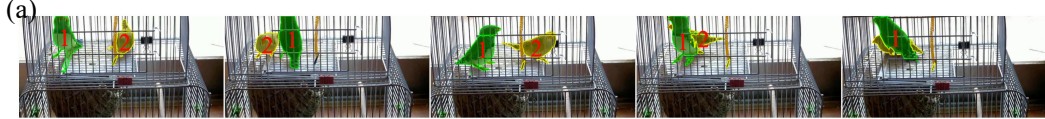

*Query:* The birds inside the cage, continuously hopping around.

*Reasoning:* <think> To determine the event 'The birds inside the cage, continuously hopping around.', I need to observe the behavior of the birds in the video. The birds appear to be moving around the cage, which suggests they are hopping. The bird labeled as '1' seems to be more active, moving around the cage more frequently than the bird labeled as '2'. The bird labeled as '2' appears to be less active and mostly stationary. </think>

<answer> Target ID: 1, 2 </answer>

(b)

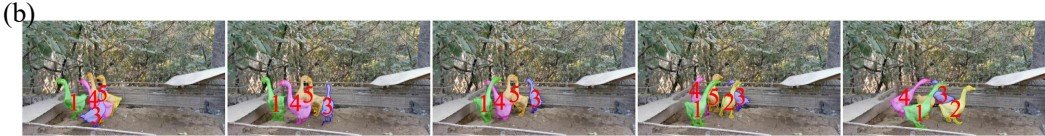

*Query:* A cluster of wandering domesticated birds.

*Reasoning:* <think> The event 'A cluster of wandering domesticated birds.' seems to occur throughout the video as the geese move around the enclosure. However, the geese appear to be more active and moving together in the middle part of the video, specifically from around 0.00 to 10.00 seconds. </think>

<answer> Target ID: 1, 2, 3, 4, 5 </answer>

(c)

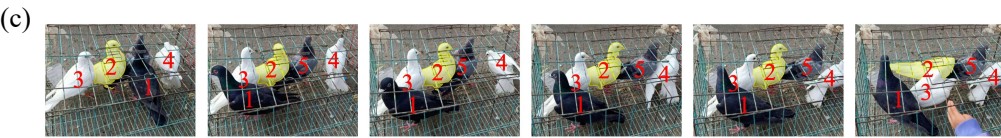

*Query:* The white pigeon that hasn't moved from its position.

*Reasoning:* <think> To determine the event 'The white pigeon that hasn't moved from its position,' I need to identify the white pigeon in the cage and observe if it remains stationary throughout the video. The white pigeon is located in the middle of the cage, and it appears to be the second pigeon from the left. It seems to be the only one that hasn't moved significantly compared to the others. </think>

<answer> Target ID: 2 </answer>

(d)

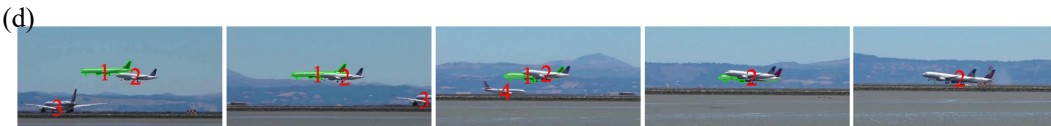

*Query:* bigger one of the two planes moving left and landing.

*Reasoning:* <think> The event 'bigger one of the two planes moving left and landing' can be observed from the moment the plane starts descending towards the runway until it touches down. The plane labeled as '1' is the bigger one among the two planes. It is positioned on the right side of the frame and moves leftward as it lands.</think>

<answer> Target ID: 1 </answer>

Figure 9: Cases for referring video object segmentation.

### A.3 MORE EXPERIMENTS

Since video temporal grounding task does not inherently require object-centric visual prompts, their addition may even obscure fine-grained video details and slightly hinder temporal accuracy in the zero-shot setting. To further verify this, we compare STVG-R1 with and without visual prompts on Charades-STA and TVGBench. As shown in Table 10, visual prompts bring a marginal decrease in performance, indicating that visual prompts are not essential for temporal-only task.

Table 10: Comparison of STVG-R1 with and without visual prompts on temporal grounding benchmarks Charades-STA and TVGBench (%). Adding visual prompts slightly affects temporal performance, showing that object-centric prompts are less critical for temporal-only tasks.

| Models | Charades-STA | | TVGBench | |
|---|---|---|---|---|
| | tIoU@0.3 | tIoU@0.5 | tIoU@0.3 | tIoU@0.5 |
| STVG-R1 w. visualprompt | 72.2 | 52.1 | 41.8 | 27.2 |
| STVG-R1 w/o. visialprompt | 73.2 | 52.5 | 42.5 | 27.4 |

### A.4 VISUALIZATIONS OF UNSEEN CATEGORY IN OBJECT DETECTOR

Figure 10 presents an example where the queried object (*fish*) is not included in the detector's taxonomy and is thus misclassified into an incorrect category (*bird*). Nevertheless, our framework assigns a consistent ID and correctly localizes the target instance. This demonstrates that category misclassification does not affect the effectiveness of our approach.

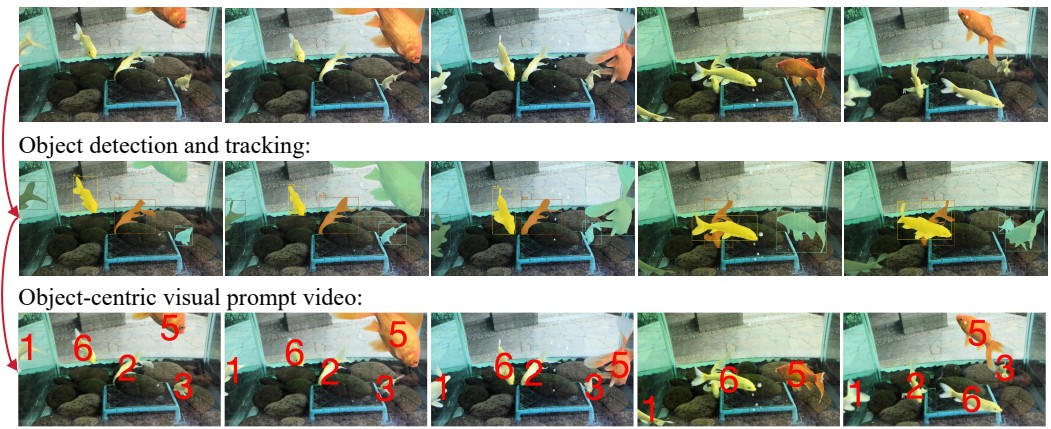

*Query: Red fish with the most pronounced up-and-down motion*

> ***Reasoning:*** <think> To identify the red fish with the most pronounced up-and-down motion, I need to observe the movement of each fish in the video. The red fish appears to be the one labeled as "5" on the right side of the tank. It seems to have a more noticeable up-and-down motion compared to the other fish. </think>
>
> <answer> Target ID: 5 </answer>

Final segmentation results:

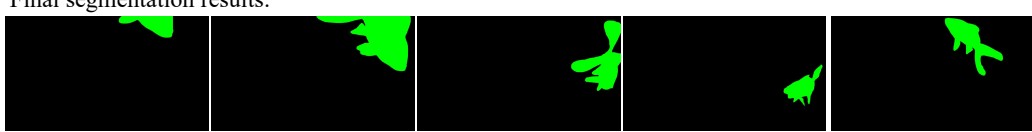

Figure 10: Visualizations of unseen category in object detector.

## A.5 ID-Repair mechanism during evaluation

To improve evaluation robustness, we introduce an ID-repair mechanism that corrects missing or inconsistent instance IDs inside the model-predicted temporal segment. Although the detector and SAM2 tracker are generally reliable, occasional ID fragmentation or missed detections can occur, leading to incomplete ID sequences. The ID-repair mechanism algorithm is provided below.

---

**Algorithm 1** ID-Repair mechanism during evaluation

---

**Require:** Predicted temporal segment $[t_s, t_e]$; detected boxes $\{B_t\}$ and IDs $\{ID_t\}$ for all frames $t$; target instance ID $ID^*$.

1: Initialize ID-correction set $A = \emptyset$
2: Let $F = \{t_s, \ldots, t_e\}$
3: **for** each frame $t \in F$ **do**
4:     **if** $ID^* \in ID_t$ **then**
5:         **continue**
6:     **end if**
7:     **Apply ID correction using** $A$
8:     **for** each $old\_id \in A$ **do**
9:         **if** $old\_id \in ID_t$ **then**
10:             Replace $old\_id$ with $ID^*$ in $ID_t$
11:         **end if**
12:     **end for**
13:     **if** $ID^* \in ID_t$ **then**
14:         **continue**
15:     **end if**
16:     Find nearest frame $t_{\text{ref}} \in F$ where $ID^* \in ID_{t_{\text{ref}}}$
17:     **if** $t_{\text{ref}}$ exists **then**
18:         Let $b_{\text{ref}}$ be the box of $ID^*$ in $B_{t_{\text{ref}}}$
19:         Compute IoU between $b_{\text{ref}}$ and each $b \in B_t$
20:         Find $b_{\text{best}}$ with the highest IoU to $b_{\text{ref}}$
21:         **if** $\text{IoU}(b_{\text{ref}}, b_{\text{best}}) \geq 0.4$ **or** $\dfrac{\text{Area}(b_{\text{ref}} \cap b_{\text{best}})}{\min(\text{Area}(b_{\text{ref}}), \text{Area}(b_{\text{best}}))} \geq 0.6$ **then**
22:             $old\_id \leftarrow$ ID of $b_{\text{best}}$
23:             Replace $old\_id$ with $ID^*$ in $ID_t$
24:             Add $old\_id$ to $A$ {persist this correction for later frames}
25:         **end if**
26:     **end if**
27:     **if** $ID^* \notin ID_t$ **then**
28:         Assign $ID^*$ to the box in $B_t$ with largest area
29:     **end if**
30: **end for**

---

## A.6 Ablation of Visual Prompting Pipeline Components

Table 11 reports ablations on the major components of the preprocessing pipeline. Periodic re-detection is essential, as removing it significantly degrades vIoU by failing to capture objects that appear after the first frame. In contrast, omitting backward tracking leads to a smaller performance drop, indicating that forward tracking alone can recover most trajectories. Thus, backward tracking can be omitted when stricter runtime efficiency is required.

Table 11: Ablation study of preprocessing components on HCSTVG-v1 test set (%).

| Method | m_tIoU | m_vIoU | vIoU@0.3 | vIoU@0.5 |
|---|---|---|---|---|
| w/o re-detection | 56.1 | 27.8 | 45.4 | 17.4 |
| w/o backward tracking | 56.8 | 28.4 | 66.2 | 37.1 |
| Full pipeline | 56.9 | 39.1 | 66.7 | 38.6 |

## A.7 VISUALIZATIONS FOR ADDITIONAL DOWNSTREAM TASKS

To further demonstrate the applicability of object-centric visual prompting beyond STVG, we provide qualitative examples for two downstream tasks: video question answering and multi-person video captioning.

**Video Question Answering.** Figure 11 shows cases where the queried entity lacks distinctive appearance cues, making it difficult for a general VLM to localize the correct subject. Without visual prompts, the model fails to answer the question. With object-centric prompts (e.g., "Person 1"), the model correctly grounds the target individual and produces the correct response.

(a) **Input the raw video.**

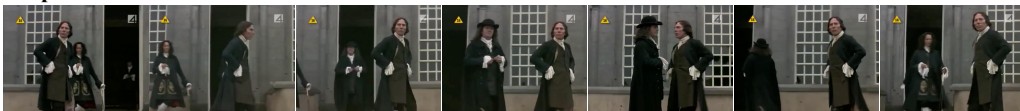

*Query:* What is the person in black doing?

> *Qwen2.5-VL-7B:* The person in black appears to be walking out of a building, followed by another individual dressed in similar attire. They both seem to be engaged in a conversation or interaction as they exit the building. The setting suggests a historical or period drama context, given their clothing and the architecture of the building.

**Input the video with visual prompt.**

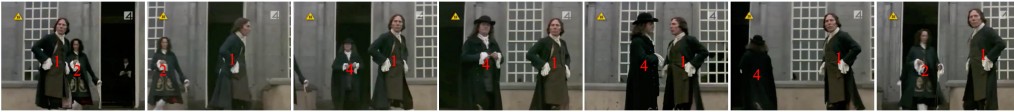

*Query:* What is the person 1 doing?

> *Qwen2.5-VL-7B:* The person with object ID 1 appears to be standing outside a building, dressed in historical attire. They seem to be observing or waiting for someone, as they have one hand on their hip and the other holding a white-gloved hand. The posture suggests a moment of pause or contemplation.

(b) **Input the raw video.**

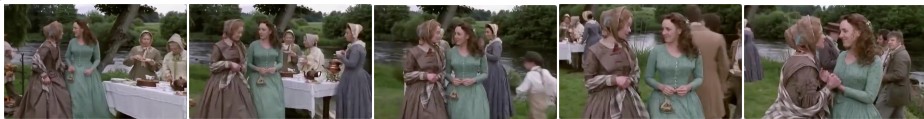

*Query:* What is the person in blue dress doing?

> *Qwen2.5-VL-7B:* The person in blue appears to be engaged in conversation with another individual, possibly sharing a moment of laughter or discussion. The setting suggests a social gathering, and the person's body language indicates they are actively participating in the interaction. The background shows other individuals dressed in period attire, suggesting this scene might be from a historical drama or reenactment.

**Input the video with visual prompt.**

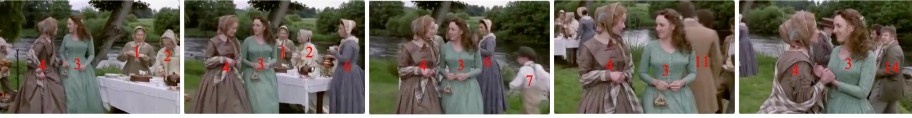

*Query:* What is the person 6 doing?

> *Qwen2.5-VL-7B:* The person with object ID 6 appears to be holding a plate and seems to be serving food or drinks from a table that is set up outdoors. The setting suggests a social gathering or a picnic, where people are enjoying refreshments.

Figure 11: A visualization of video question answering with and without visual prompting.

**Multi-person Video Captioning.** Figure 12 illustrates how visual prompts benefit video captioning. Without prompts, the model tends to generate a single global caption with ambiguous references. With instance IDs, the model produces entity-specific descriptions for each individual.

**Input the raw video.**

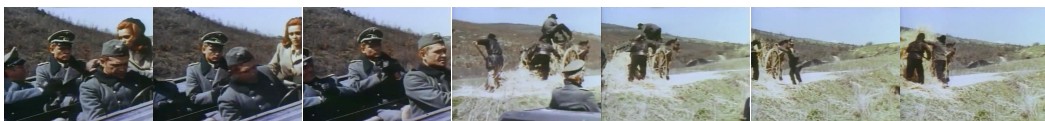

*Query:* Describe each person in this video.

> *Qwen2.5-VL-7B:* The video depicts a group of individuals dressed in military uniforms, likely from a historical or war-themed setting. The uniforms are dark-colored with distinct insignia and caps, suggesting they might be part of a specific military unit or organization. **One individual is seen** wearing a lighter-colored uniform, possibly indicating a different rank or role within the group. The scene transitions to show a horse-drawn cart moving through a rural landscape. The cart is being pulled by two horses, and there are people around it, **some standing** and **others walking alongside**. The environment appears to be a countryside with open fields and hills in the background, under a clear sky. The overall atmosphere suggests a historical or period piece, possibly depicting a military or civilian scene from a past era.

**Input the video with visual prompt.**

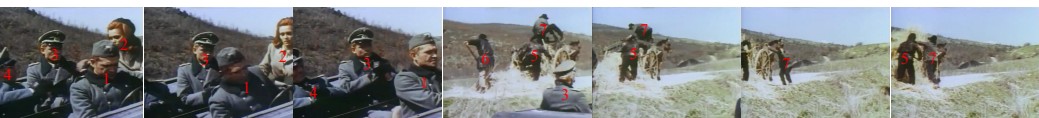

*Query:* Describe each person in this video.

> *Qwen2.5-VL-7B:* Here is a description of each person in the video:
> 1. **Object ID 1, 3, 4**: This person is seated in the back of a vehicle, wearing a military uniform and a cap. They appear to be looking forward.
> 2. **Object ID 2**: This person is standing next to a vehicle, wearing a military uniform and a cap. He appears to be looking forward.
> 3. **Object ID 5**: This person is standing next to a horse-drawn cart, wearing a dark outfit and appears to be holding onto the cart.
> 4. **Object ID 6**: This person is standing next to a horse-drawn cart, wearing a dark outfit and appears to be holding onto the cart.
> 5. **Object ID 7**: This person is standing on top of a horse-drawn cart, wearing a dark outfit. He jumped down from the horse-drawn cart.
> The video shows these individuals in various positions within and around a horse-drawn cart, with some of them wearing military uniforms and caps. The setting appears to be outdoors, possibly in a rural or mountainous area.

Figure 12: A visualization of multi-person video captioning enhanced by visual prompting.

### A.8 ADDITIONAL QUALITATIVE RESULTS ON DIVERSE VIDEO DOMAINS

To further assess the generalization ability of our approach, we conduct qualitative evaluations on videos drawn from two distinct domains: ego-centric videos from Ego4D Grauman et al. (2022) and movie videos from Grounded-VIDSitu Khan et al. (2022). As shown in Figures 13 and 14, our method maintains stable instance grounding performance despite challenges such as rapid camera motion, complex scene composition, and multiple interacting entities.

(a)

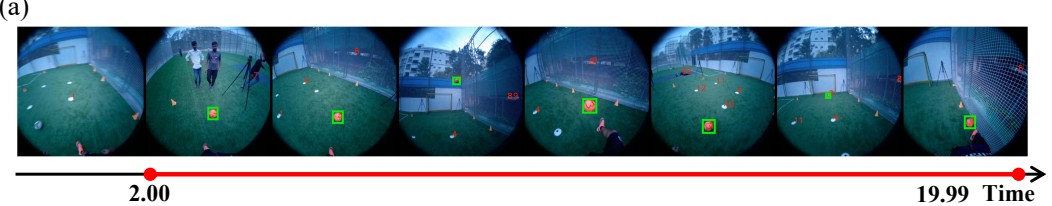

**2.00**                                                                                          **19.99  Time**

*Query:* A person kicks a soccer ball.

> *Reasoning:* <think> The video shows a person practicing soccer drills on an indoor field. The person is seen kicking a soccer ball towards a goalpost. The ball is visible in the air during the kick and lands near the goalpost. The ball is marked with the number 3. </think>
>
> <answer> Target ID: 3, Time range: 2.00 to 19.99 </answer>

(b)

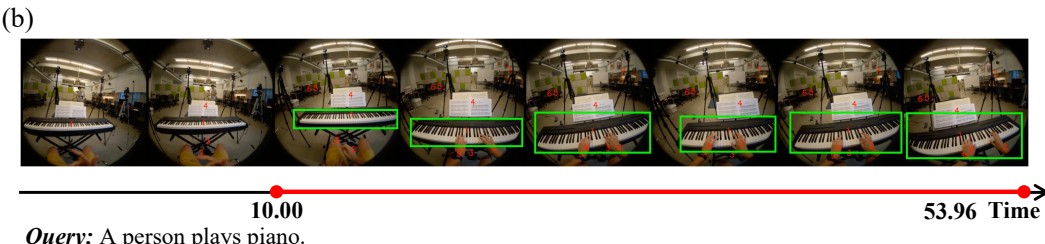

**10.00**                                                                                     **53.96  Time**

*Query:* A person plays piano.

> *Reasoning:* <think> The event "A piano plays piano." occurs throughout the entire video. The piano is consistently visible in the foreground, and it is the main focus of the video. The piano's keys are being played by the person in the video, indicating that the piano is actively being used. </think>
>
> <answer> Target ID: 1, Time range: 10.00 to 53.96 </answer>

Figure 13: Qualitative results on ego-centric videos from Ego4D Grauman et al. (2022).

(a)

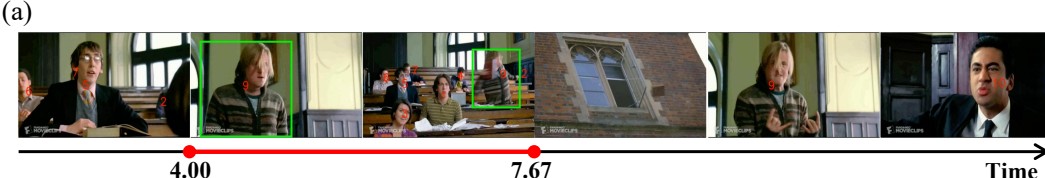

**Query:** A man stand up angrily and throw the book out the window.

*Reasoning:* <think> The event "A man stand up angrily and throw the book out the window." occurs when the man in the striped sweater stands up and throws the book out of the window. This action happens after he has been speaking and gesturing, as indicated by his body language and facial expressions. </think>

<answer> Target ID: 9, Time range: 4.00 to 7.67 </answer>

(b)

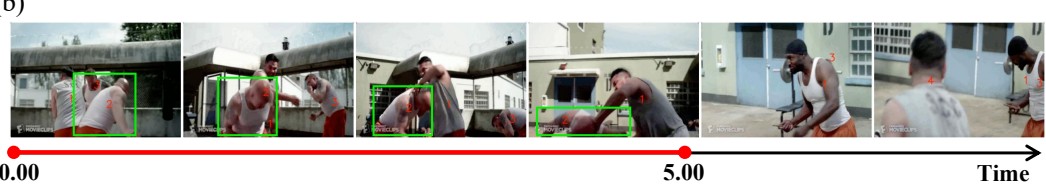

**Query:** A bald man is beaten.

*Reasoning:* <think> The event "A bald man is beaten." occurs when the bald man is engaged in a physical altercation with another individual. The bald man is identified by his white tank top and orange shorts. The sequence of frames shows him being pushed and punched by other individuals, indicating that he is being beaten. </think>

<answer> Target ID: 2, Time range: 0.00 to 5.00 </answer>

Figure 14: Qualitative results on movie-style videos from Grounded-VIDSitu Khan et al. (2022).

## A.9 REWARD ABLATION

We analyze the effect of the format reward ($r_f$) and a coupled reward variant. All experiments are conducted on HCSTVG-v1.

**Effect of removing $r_f$.** The training curves (Fig. 15) indicate that removing $r_f$ produces nearly identical optimization dynamics. Since Qwen2.5-VL naturally supports <think> and <answer> tokens, the marginal benefit of the simple format constraint is limited.

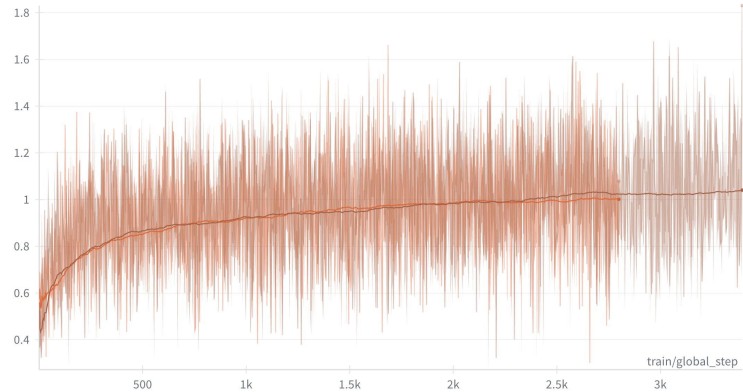

Figure 15: Training curves with and without the format reward $r_f$. The brown curve corresponds to the full reward setting, while the orange curve corresponds to the model trained without $r_f$.

**Coupled reward variant.** The training curves of coupled spatial reward and decoupled spatial reward are as shown in Fig. 16, which indicate that both formulations exhibit nearly identical convergence behavior.

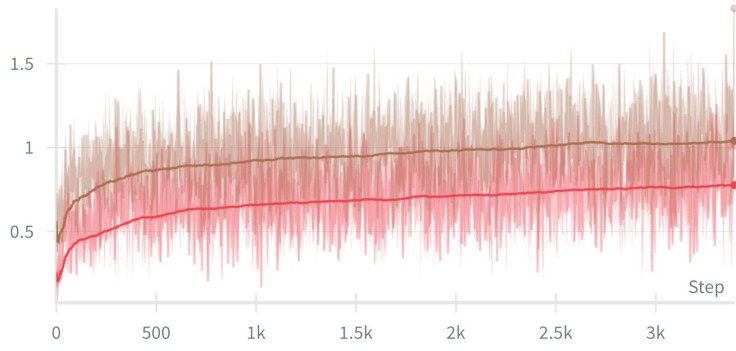

Figure 16: Training curves comparing the decoupled brown reward and the coupled red reward.

## A.10 THE USE OF LARGE LANGUAGE MODELS (LLMS)

We employed large language models for language polishing to improve the clarity and readability of the manuscript. Specifically, LLMs were used to refine grammar, adjust sentence structure, and enhance overall flow, without altering the technical details and experimental results.

