# OpenReview forum: "STVG-R1: Incentivizing Instance-Level Reasoning and Grounding in Videos via Reinforcement Learning"
_ICLR.cc/2026/Conference — ICLR 2026 Poster_

### Official Review · Reviewer_enhd · 2025-10-24

**Soundness:** 2
**Presentation:** 3
**Contribution:** 2
**Rating:** 4
**Confidence:** 3

**Summary:**

This paper proposes a new visual prompt paradigm without alignment coordinates, which transforms frame by frame coordinate prediction into instance level ID recognition. For the first time, reinforcement learning has been introduced into spatial-temporal video grounding, and task driven reward joint optimization of temporal, spatial, and format constraints has been designed. In multiple benchmark tests, refreshing SOTA resulted in a 20.9% improvement in mIoU for HCSTVG-v2.

**Strengths:**

1. The writing of the paper is clear, and the illustrations in the intro section effectively explain the core contribution points of this paper. In addition, the drawing of Figures 2 and 3 is also quite intuitive.

2. The problem studied in this article (cross modal alignment) is one of the core issues in the field of video grounding, and the alignment effect between the spatial-temporal dimensions directly determines the accuracy of grounding in these two dimensions.

3. The experiment used four commonly used indicators and demonstrated the accuracy of the model on the dataset used.

**Weaknesses:**

1. The last sentence of the second paragraph of the intro should be a description of the core idea, but this sentence is not clear. What does' a compact and interpretable formulation 'refer to.

2. Insufficient contribution in model design, only introducing pre-segmentation and reinforcement learning for video objects, lacking new method design for this problem.

3. The commonly used datasets in the field of Video Grounding include VidSTG and HC-STVG. This article only uses HC-STVG, and the effect on VidSTG is unknown, especially the results of ablation experiments.

**Questions:**

See [Weaknesses].

---

> ### Author Response · Authors · 2025-11-21
> **Response To Reviewer enhd [1/2]**
>
> Thanks for your insightful and thorough review. We will address your concerns one by one.
>
> > **W1.** The last sentence of the second paragraph of the intro should be a description of the core idea, but this sentence is not clear. What does "a compact and interpretable formulation" refer to.
>
> **R1.** We thank the reviewer for pointing out the lack of clarity. The phrase "a compact and interpretable formulation" refers to the transformation of the STVG task from dense, per-frame spatial predictions into a discrete instance-level ID selection problem. **Traditional STVG systems require the model to output a sequence of bounding boxes,** which are high-dimensional, difficult to interpret, and do not explicitly expose the model’s reasoning process. In contrast, **our object-centric visual prompting pipeline reduces the decision space to choosing from a finite set of instance IDs.** This leads to a much more compact representation (from each frame's bounding box to an instance ID) and makes the model’s output directly interpretable (e.g., selecting "ID = 3," corresponding to a clearly visualized object in the video). Furthermore, this formulation enables the model to express the reasoning process in a structured interpretable manner, such as "ID 1 is doing X" or "ID 3 matches the described action," which makes it naturally compatible with GRPO reasoning process. We have updated the manuscript in Introduction section to clarify this point explicitly:
>
> "Motivated by these challenges and prior attempts, we propose the following insight: by reformulating STVG from dense per-frame coordinate prediction into a compact and interpretable instance-level ID selection problem, it becomes possible to mitigate visual–textual misalignment and enhance generalization."
>
> > **W2.** Insufficient contribution in model design, only introducing pre-segmentation and reinforcement learning for video objects, lacking new method design for this problem.
>
> **R2.** We thank the reviewer for the thoughtful comment, and we acknowledge that our method does not introduce new architectural modules. However, we fully disagree with the conclusion that the work lacks new method design. **The objective of this work is not to propose a new detection or segmentation architecture, but to introduce a new paradigm for spatio-temporal video grounding (STVG).** A key methodological contribution of our work is the introduction of object-centric visual prompting in STVG, which enables general VLMs to achieve competitive or superior performance without any task-specific learnable components or large auxiliary datasets. **Using a general-purpose VLM to address complex video understanding tasks, rather than building task-specific architectures, reflects an important developmental trend in modern model research.**
>
> While future work may explore integrating the preprocessing components into a unified model, such refinements are beyond the scope of the conceptual contribution we aim to highlight in this paper.

---

> > ### Author Response · Authors · 2025-11-21
> > **Response To Reviewer enhd [2/2]**
> >
> > > **W3.** The commonly used datasets in the field of Video Grounding include VidSTG and HC-STVG. This article only uses HC-STVG, and the effect on VidSTG is unknown, especially the results of ablation experiments.
> >
> > **R3.** Thank you for raising this point. In addition to HC-STVG, we also evaluate our method on **ST-Align, a dataset constructed on VidSTG** that refines the textual descriptions using GPT-4-turbo to enable clearer and more unambiguous referential expressions. Moreover, **ST-Align includes both spatial-temporal video grounding and spatial video grounding tasks,** providing a more comprehensive evaluation of the model’s video understanding capability.
> >
> > We initially experimented on the VidSTG dataset as well. However, we found that a non-negligible portion of the samples with unspecified text, where the textual description is compatible with multiple instances in the video, making reliable evaluation difficult. For example, queries such as "there is a ski beneath an adult in the wild" appear in crowded ski, yet only one instance is labeled as the target. These ambiguous samples introduce substantial noise in model performance evaluation.
> >
> > We present our VidSTG results below.
> >
> > Table A. Comparison with existing state-of-the-art models on 30% randomly sampled VidSTG test set (%) with declarative sentences. The results of GroundingGPT-7B are reported from SpaceVLLM, while those of Qwen2.5-VL-7B and Qwen3-VL-8B are generated by our experiments.
> >
> > | Models | m_tIoU | m_vIoU | vIoU\@0.3 | vIoU\@0.5 |
> > |--------|--------|--------|-----------|-----------|
> > | TubeDETR | 48.1 | 30.4 | 42.5 | 28.2 |
> > | STVGFormer | – | 33.7 | 47.2 | 32.8 |
> > | CG-STVG | 51.4 | 34.0 | 47.7 | 33.1 |
> > | TA-STVG | 51.7 | 34.4 | 48.2 | 33.5 |
> > |--------|--------|--------|-----------|-----------|
> > | GroundingGPT-7B | 15.5 | 12.3 | 13.2 | 4.1 |
> > | SpaceVLLM-7B | 47.7 | 27.4 | 39.1 | 26.2 |
> > | Qwen2.5-VL-7B | 38.3 | 9.3 | 13.0 | 6.0 |
> > | **+ VisualPrompt** | 38.4 | 14.4 | 18.6 | 8.6 |
> > | Qwen3-VL-8B | 44.7 | 10.7 | 14.9 | 7.6 |
> > | **+ VisualPrompt** | 44.7 | 15.3 | 19.0 | 9.9 |
> > | **STVG-R1** | 49.7 | 30.5 | 43.9 | 27.5 |
> >
> > Table B. Comparison with existing state-of-the-art models on 30% randomly sampled VidSTG test set (%) with interrogative sentences. The results of GroundingGPT-7B are reported from SpaceVLLM, while those of Qwen2.5-VL-7B and Qwen3-VL-8B are generated by our experiments.
> >
> > | Models | m_tIoU | m_vIoU | vIoU\@0.3 | vIoU\@0.5 |
> > |--------|--------|--------|-----------|-----------|
> > | TubeDETR | 46.9 | 25.7 | 35.7 | 23.2 |
> > | STVGFormer | – | 28.5 | 39.9 | 26.2 |
> > | CG-STVG | 49.9 | 29.0 | 40.5 | 27.5 |
> > | TA-STVG | 50.2 | 29.5 | 41.5 | 28.0 |
> > |--------|--------|--------|-----------|-----------|
> > | GroundingGPT-7B | 11.9 | 8.7 | 9.6 | 2.9 |
> > | SpaceVLLM-7B | 48.5 | 25.4 | 35.9 | 22.2 |
> > | Qwen2.5-VL-7B | 35.8 | 5.7 | 7.0 | 2.9 |
> > | **+ VisualPrompt** | 37.0 | 12.6 | 16.5 | 7.1 |
> > | Qwen3-VL-8B |43.3 | 8.0 | 10.9 | 4.8 |
> > | **+ VisualPrompt** | 43.5 | 15.3 | 19.8 | 11.2 |
> > | **STVG-R1** | 48.1 | 26.5 | 37.0 | 22.3 |
> >
> > The results show that, compared with VLM-based approaches such as GroundingGPT-7B and SpaceVLLM-7B, our STVG-R1 achieves strong and consistent improvements across all metrics when evaluated with declarative sentences. Moreover, under interrogative sentence evaluation, STVG-R1 also achieves performance comparable to the strongest VLM-based baseline (SpaceVLLM-7B), further validating the effectiveness of our approach across different query types. Although the performance does not yet surpass the best specialized STVG models on this dataset, the results clearly indicate that a general-purpose VLM equipped with visual prompting and GRPO possesses substantially stronger cross-dataset generalization ability.

---

### Official Review · Reviewer_bVvE · 2025-11-01

**Soundness:** 3
**Presentation:** 3
**Contribution:** 2
**Rating:** 4
**Confidence:** 4

**Summary:**

Brief Summary: The paper tackles the task of video spatio-temporal localization where given a video and a corresponding query, the model should output the bounding boxes + temporal timestamps. The authors propose visual prompting using a combination of yolov12 detector (for object detection) + sam2 (for visual tracking) and overlaying the information on the image itself and providing particular object ids which are also overlayed. Given the visual-prompted video, a Qwen2.5-VL model is trained via GRPO to predict the target id (which has associated bounding box) + time-range. Experiments on spatio-temporal datasets like HCSTVGv1, v2, and temporal grounding like charades-sta show that proposed STVG-R1 outperforms competitive baselines.

**Strengths:**

Pros:

1. The paper poses a nice application of combining spatio-temporal understanding with VLMs. Spatio-temporal understanding is an important sub-topic in video understanding and how to best leverage VLMs for this task is a well motivated problem.

2. The proposed method is conceptually simple in re-using existing detection and tracking pipelines to utilize VLMs inherent understanding of vision without additional tokens or requiring VLM to do additional bounding box predictions.

3. Authors provide visualization (in appendix A.2) and ablation on various prompt designs. In particular, Table 8 is very interesting, that pure SFT without GRPO training leads to worse results.

**Weaknesses:**

Cons:

1. It seems the absolute improvement over previous baselines is marginal? For instance on hcstvg-v1, performance matches with space-vllm and on v2, it is slightly improved over TA-STVG. On ST-Align, it is same as LLava-ST-7B.

2. The core novelty is slightly limited, the paper suggests doing visual-prompting + grpo training works. This is good to know, but unclear what are the main challenges here.

3. One issue with the visual prompting (assuming the visualization at face value), it is unclear how the approach would tackle things with text (OCR). If you overlay a color bounding box, the text is completely lost. So the model cannot answer questions like "when did the person look at <some_text>".

4. The model is essentially restricted to detection quality and classes of yolov12 and the quality of sam2. As such, there is no direct way to leverage VLMs internal association capability. Further dependence on separate models would lead to worse inference times requiring heavy video encoding/decoding.

5. Table 9 in Appendix A.3 seems to suggest direct visual prompting is in fact worse? That seems like a major drawback?

6. (Minor) It would be interesting to see results on more diverse videos, such as some ego-centric datasets (such as ego4d) or movie datasets (such as grounded-vidsitu [Ref1]).

7. The main comparison to baselines is somewhat unfair. The proposed model is able to leverage external models for tracking while baseline models need to do the prediction on their own? I could be missing something obvious here.

8. (Minor) I am slightly confused on the r_s reward, why is it not simply 3d-iou?

9. Authors should show additional downstream tasks which gain from such visual prompting.

---

[Ref1]: Khan, Zeeshan, C. V. Jawahar, and Makarand Tapaswi. "Grounded video situation recognition." Advances in Neural Information Processing Systems 35 (2022): 8199-8210.

---

Overall Rating: 4/10
The paper proposes grpo RL training with appropriate visual prompting for spatio-temporal video grounding. The scope however is somewhat narrow and not shown to be applicable for other video understanding tasks, and seems to slightly degrade temporal grounding. The proposed visual prompting itself might interfere with ocr reasoning, and comparison to baselines is not strictly fair.

**Questions:**

Q1. In general, it is advisable to do an initial cold-start before GRPO, but I don't see any references on that in the paper. Did the authors try it and didn't give good results?

---

> ### Author Response · Authors · 2025-11-21
> **Response To Reviewer bVvE [1/3]**
>
> Thanks for your insightful and thorough review. We will address your concerns one by one.
>
> > **W1.** It seems the absolute improvement over previous baselines is marginal? For instance on hcstvg-v1, performance matches with space-vllm and on v2, it is slightly improved over TA-STVG. On ST-Align, it is same as LLava-ST-7B.
>
> **R1.** We would like to clarify that the improvements are not marginal. **On HCSTVG-v2** benchmark, our STVG-R1 surpasses the strongest existing baseline TA-STVG with gains of +1.6% m_tIoU, +2.0% vIoU\@0.3, and +2.1% vIoU\@0.5, which constitute **a substantial and consistent improvement** over prior methods. For HCSTVG-v1 and ST-Align benchmarks, while our results are numerically comparable to the results of SpaceVLLM-7B and LLava-ST-7B, it is important to note that both SpaceVLLM-7B and LLaVA-ST-7B are trained with very large external datasets: **SpaceVLLM-7B uses approximately 480K instance-level annotations, and LLaVA-ST-7B is trained on roughly 4.3M samples**. In contrast, STVG-R1 is trained only on the standard STVG training sets (HC-STVG and VidSTG), which together contain about **27K samples**. Achieving comparable performance under this strictly limited data indicates that our method is **highly data-efficient**. Finally, unlike prior baselines that are designed for a single task, STVG-R1 uses **one unified model** to achieve SOTA or highly competitive performance across spatial–temporal video grounding, spatial video grounding, video temporal grounding, and multi-object segmentation tasks, demonstrating stronger generalization rather than isolated gains.
>
> > **W2.** The core novelty is slightly limited, the paper suggests doing visual-prompting + grpo training works. This is good to know, but unclear what are the main challenges here.
>
> **R2.** We thank the reviewer for the valuable comment. Existing VLM-based methods on STVG typically stack trainable alignment modules or decoders on top of a general VLM, essentially performing coordinates regression. In contrast, our approach reformulates dense per-frame coordinates prediction into a compact instance-level ID recognition problem. With object-centric visual prompts, a general VLM can achieve competitive or superior performance on STVG.
>
> The primary technical challenge lies in **robustly handling the noise introduced by the external detector and segmenter**. Their outputs inevitably contain missed detections or inconsistent ID assignments. To make our model robust to these errors, our pipeline is designed to suppress this noise at three stages:
>
> 1. **Preprocessing:** Periodic re-detection combined with bidirectional SAM2 tracking recovers objects that momentarily drop out or detection failed.
> 2. **Training:** Frame-level supervision is obtained via IoU matching and then aggregated through majority voting across the video, making the final label robust to local frame-level ID errors.
> 3. **Evaluation:** When the predicted ID is missing in part of the model-predicted temporal interval, an ID-repair mechanism propagates the nearest valid ID with IoU-based matching (detailed in Algorithm 1 of the updated manuscript in Appendix A.5), further mitigating the effect of local tracking or ReID errors.
>
> Together, these components ensure that detector/segmenter noise has minimal influence on both training and inference process, enabling stable and accurate performance.
>
> > **W3.** One issue with the visual prompting (assuming the visualization at face value), it is unclear how the approach would tackle things with text (OCR). If you overlay a color bounding box, the text is completely lost. So the model cannot answer questions like "when did the person look at [object Object]".
>
> **R3.** We clarify that the **only visual prompt inserted into the video is the small numeric ID at the object centroid**, as shown in Figures 3, 11 and 12 in the updated manuscript. **No color bounding boxes** are added during inference, and therefore no large-area occlusion occurs. The color bounding boxes shown in Figures 4 and 8 are only for visualization and are not the model input.
>
> Regarding OCR-related scenarios, the benchmarks used in our experiments do not contain text-heavy queries, and our method has not been evaluated on text-heavy data. We acknowledge that placing the ID directly at the centroid may interfere with nearby text. **However, this is not a limitation of the approach itself, and a simple placement choice can easily mitigate the issue.** For example, one can detect text regions using a lightweight OCR detector and place the ID at a non-text corner of the object mask (e.g., upper-left or upper-right) instead of the centroid. In our future open-source release, we plan to include an option that automatically adjusts ID placement to avoid text regions, making the pipeline robust for OCR-heavy scenarios.

---

> ### Author Response · Authors · 2025-11-21
> **Response To Reviewer bVvE [2/3]**
>
> > **W4.** The model is essentially restricted to detection quality and classes of yolov12 and the quality of sam2. As such, there is no direct way to leverage VLMs internal association capability. Further dependence on separate models would lead to worse inference times requiring heavy video encoding/decoding.
>
> **R4.** We acknowledge the concern. However, Table 6 in the manuscript shows that our pipeline provides **a strong upper bound**, reaching vIoU\@0.3 of up to 95%. To further understand the remaining errors, we manually inspected the 100 lowest high upper bound samples. **Only about 25% of the failures were caused by detector or segmenter limitations** (mostly very small objects). The remaining 75\% were due to noisy benchmark annotations, many of which were produced by the earlier tracking-based annotation tools.
>
> Regarding the comment on internal association capability, the core objective of STVG is to identify the correct instance and determine its temporal span. Our design leverages the VLM’s strong reasoning ability for this decision-making step, which aligns with the task’s requirements. We intentionally avoid relying on the VLM’s internal association capability to perform implicit tracking, as such behavior is known to be unstable.
>
> Finally, we evaluated runtime on the 20s duration HC-STVG videos using a single RTX 4090, averaging results over 100 samples. Inference with our STVG-R1 model takes 81.47s per video. The preprocessing "visual prompting" pipeline takes 115.24s at the original video FPS, with SAM2 contributing roughly 87% of total time. Reducing the input to 5 FPS decreases preprocessing time to 38.10s while preserving comparable accuracy, as the m_vIoU decreases only marginally from 39.7% to 38.9%. Under this setting, preprocessing becomes approximately half as expensive as VLM inference.
>
> > **W5.** Table 9 in Appendix A.3 seems to suggest direct visual prompting is in fact worse? That seems like a major drawback?
>
> **R5.** We thank the reviewer for the observation. Table 9 in Appendix A.3 reports results on video **only-temporal** grounding (VTG), for which visual prompts are inherently unnecessary. In this context, adding visual prompts introduces a small amount of visual alteration that may slightly obscure motion cues, leading to **a very minor performance change:** from 52.5% to 52.1% on Charades-STA and from 27.4% to 27.2% on TVGBench at tIoU\@0.5. These differences are within normal variation and confirm our claim in the paper that **visual prompts are not essential for temporal-only task.** The impact is extremely limited.
>
> Moreover, the effect is **not consistently negative** across models. On HCSTVG-v1/v2 temporal evaluation in Table 1, InterVL3-8B and Qwen3-VL-8B exhibit slight improvements, while Qwen2.5-VL-7B and Qwen2.5-VL-72B show small decreases. **This reflects model fluctuations rather than a methodological weakness.**
>
> > **W6.** (Minor) It would be interesting to see results on more diverse videos, such as some ego-centric datasets (such as ego4d) or movie datasets (such as grounded-vidsitu [Ref1]).
>
> **R6.** Thank you for the suggestion. We conducted experiments on a small subset of ego4d and grounded-vidsitu videos to assess the generalization ability of our model on more diverse video domains. **Several visulization cases are included in Appendix A.8 in the updated manuscript.** Figure 13 shows results on ego-centric videos, demonstrating that our object-centric prompting remains effective under strong camera motion. Figure 14 presents examples from movie videos, where our method is also able to localize the correct instance in complex scene. These qualitative results indicate that the proposed method generalizes well beyond the benchmarks used in the main paper.
>
> > **W7.** The main comparison to baselines is somewhat unfair. The proposed model is able to leverage external models for tracking while baseline models need to do the prediction on their own? I could be missing something obvious here.
>
> **R7.** Yes, it is true that previous baselines do not use external models, whereas our method does rely on external models for tracking. However, this does not create an unfair comparison. Prior STVG baselines depend on learnable task-specific decoders, additional tokens, or auxiliary alignment modules, whereas our framework replaces these learned components with a training-free visual prompt generation pipeline. This reflects a paradigm-level shift that a general VLM equipped with object-centric visual prompt can achieve competitive or superior performance on STVG, without the large auxiliary datasets or heavily engineered architectures. Fundamentally, **both our method and prior approaches add external components on top of a general VLM, so the comparison is not unfair.**

---

> > ### Author Response · Authors · 2025-11-21
> > **Response To Reviewer bVvE [3/3]**
> >
> > > **W8.** (Minor) I am slightly confused on the r\_s reward, why is it not simply 3d-iou?
> >
> > **R8.** We thank the reviewer for the insightful question. We did experiment with a continuous meanvIoU  (vIoU averaged over frames) as the spatial reward, but it performed worse than our sparse reward design. Tables below show the comparison on HCSTVG-v2.
> >
> > Table A: Different steps evaluation with **continuous vIoU spatial reward** on HCSTVG-v2 val set (%) with 2048 tokens, 1/3 filtering threshold and 2FPS.
> >
> > | steps | m_tIoU | m_vIoU | vIoU\@0.3 | vIoU\@0.5 |
> > |-------|--------|--------|-----------|-----------|
> > | 2200  | 59.7   | 38.3   | 64.9      | 35.6      |
> > | 2600  | 58.6   | 37.8   | 64.1      | 33.0      |
> > | 3000  | 61.6   | 39.9   | 67.1      | 37.4      |
> > | 3395  | 60.2   | 38.9   | 66.6      | 36.1      |
> >
> > Table B: Different steps evaluation with **sparse spatial reward** on HCSTVG-v2 val set (%) with 2048 tokens, 1/3 filtering threshold and 2FPS.
> >
> > | steps | m_tIoU | m_vIoU | vIoU\@0.3 | vIoU\@0.5 |
> > |-------|--------|--------|-----------|-----------|
> > | 2200  | 59.8   | 38.7   | 65.4      | 35.3      |
> > | 2600  | 60.8   | 39.7   | 67.1      | 37.7      |
> > | 3000  | 61.5   | 40.1   | 67.9      | 38.5      |
> > | 3395  | 62.0   | 40.2   | 67.8      | 38.8      |
> >
> > **The continuous reward does not provide better performance, and in fact introduces instability during RL training.** We summarize the reasons as follows:
> >
> > 1. In our paradigm, the binary signal better matches our instance-level formulation, where the objective is to **choose the most correct identity rather than a relatively correct one.** A continuous vIoU reward blurs this objective.
> > 2. In our framework, the VLM does not predict bounding boxes or masks. All geometric coordinates information is produced by the offline "YOLO+SAM2" preprocessing pipeline. **A continuous reward introduces noise that are unrelated to the VLM’s decision,** undermining the stability of the RL optimization.
> >
> > > **W9.** Authors should show additional downstream tasks which gain from such visual prompting.
> >
> > **R9.** We agree with the reviewer that object-centric visual prompting should extend to additional downstream tasks. For example, in **video captioning**, explicit instance IDs allow the model to describe the actions of each person in a video rather than producing a single global summary. In **video QA**, when a query refers to an instance without distinctive appearance cues, VLMs often struggle to identify the correct person, which leads to incorrect or ambiguous answers.
> >
> > We include two qualitative examples in **Appendix A.7 Visualizations for Additional Downstream Tasks** in the updated manuscript to show how visual prompting benefits video QA and video captioning tasks.
> >
> > > **Q1.** In general, it is advisable to do an initial cold-start before GRPO, but I don't see any references on that in the paper. Did the authors try it and didn't give good results?
> >
> > **RQ1.** A cold-start stage is typically introduced to provide supervised examples of step-by-step reasoning before reinforcement learning. However, this stage **requires a supervised reasoning dataset.** Such data does not exist for STVG, making a meaningful cold-start difficult to construct.
> >
> > Moreover, modern VLMs such as Qwen2.5-VL-7B already exhibit strong inherent reasoning capabilities, and several recent studies [Ref1] have show that cold-start often **brings limited benefit for models of this scale.** In addition, the reasoning required for STVG is relatively straightforward (e.g., identifying the correct instance and its temporal span), rather than involving multi-step symbolic reasoning, so the impact of a cold-start stage is expected to be small. Given these considerations, we directly applied GRPO without a cold-start stage, and **in our experiments it remains stable and effective.**
> >
> > [Ref1]: Wang, Ye, Boshen Xu, Zihao Yue, Zihan Xiao, Ziheng Wang, Liang Zhang, Dingyi Yang, Wenxuan Wang, and Qin Jin. "TimeZero: Temporal video grounding with reasoning-guided LVLM." arXiv e-prints, arXiv-2503.

---

> ### Comment · Reviewer_bVvE · 2025-11-26
>
> I thank the authors for the detailed rebuttal. Please find my comments below:
>
> 1. I don't think data efficiency is a core point given STVG-R1 utilized sam2 and yolov2 which are themselves sota models trained on a large sample set. Ideally, authors should show that using more data similar in size to SpaceVLLM-7B or LLaVA-ST-7B leads to comparable improvement.
>
> 2. Requiring OCR detection as a separate pipeline seems like a significant issue to me.
>
> 3. Appreciate response to W4, it seems the pre-processing is indeed a significant bottleneck.
>
> 4. It is still quite surprising that improvement in spatial reasoning is not leading to improvement in temporal reasoning. It could potentially be due to underlying test dataset quality itself. Not sure.
>
> 5. Additional experiments on comparison with direct 3d-iou is valuable. Appreciate it.
>
> Overall, I am inclined to keep my score as is.

---

> > ### Author Response · Authors · 2025-12-02
> > **Response To Reviewer bVvE**
> >
> > Thank you again for the careful reading of our rebuttal and for the additional comments. We sincerely appreciate the time and expertise you dedicated in our work. Below we provide clarifications on the remaining concerns.
> >
> > **1. On the "data-efficiency" point**
> >
> > We fully agree that SAM2 and YOLOv12 are trained on large detection/segmentation datasets. However, the comparison in our paper (SpaceVLLM-7B and LLaVA-ST-7B) heavily rely on large amounts of self-collected STVG-specific annotations, not generic vision data. and training with SAM2-scale data would be:
> >
> > * **computationally infeasible** under our available compute budget,”
> > * **model-specific,** meaning every new VLM architecture would require re-training on a massive task-specific dataset.
> >
> > In contrast, STVG-R1 equips any off-the-shelf VLM with STVG capability using only the standard STVG datasets (~27K samples). We view this as a meaningful contribution toward scalable and model-agnostic STVG.
> >
> > **2. On OCR concerns.**
> >
> > **(1) Visual prompts do not materially degrade OCR performance.**
> >
> > To empirically verify this, we evaluated Qwen2.5-VL-7B with and without visual prompts on MME-VideoOCR, which contains 10 OCR-centric tasks. As shown in Table B, the differences are consistently small, even in tasks most sensitive to text occlusion (TR). Several tasks (VTQA, TG) improve slightly, likely because the small numeric marker strengthens attention to the referenced object.
> >
> > Table B: Evaluation results on MME-VideoOCR. "TR" denotes Text Recognition, "VTQA" Visual Text QA, "TG" Text Grounding, "AR" Attribute Recognition, "CDT" Change Detection & Tracking, "STP" Special Text Parsing, "CFTU" Cross-Frame Text Understanding, "TBR" Text-Based Reasoning, "TBVU" Text-Based Video Understanding, and "RVT" Robust Video Testing.
> >
> > | Method | TR | VTQA | TG | AR | CDT | STP | CFTU | TBR | TBVU | RVT | Total |
> > |-------|--------|--------|-----------|-----------|-----------|-----------|-----------|-----------|-----------|-----------|-----------|
> > | Qwen2.5-VL-7B  | **69.6%** | 76.4% | 63.2% | **69.8%** | 45.7% | **64.4%** | **22.7%** | 54.7% | **38.2%** | **79.0%** | **59.4%** |
> > | **+ VisualPrompt**  | 69.3% | **77.4%** | **65.3%** | 68.5% | **46.2%** | 64.1% | **22.7%** | **54.8%** | **38.2%** | 78.8% | 58.9% |
> >
> > **(2) OCR-aware placement is optional.**
> >
> > The lightweight OCR detector proposed in our rebuttal is only an optional safeguard for researchers working with extremely text-dense videos. Our standard pipeline does not require OCR detection, and empirical results show it is unnecessary for typical benchmarks. This option is provided out of scientific responsibility, not as a core dependency.
> >
> > **3. On "pre-processing as a bottleneck."**
> >
> > There seems to be a misunderstanding around the 25% mentioned in R4 in our rebuttal. The 25% does not mean preprocessing causes 25% of all errors. The 25% refers only to the subset of samples with low upper bound. In fact, Table 6 in the manuscript shows the upper bound is extremely high (vIoU\@0.3 = 94.6% at 1/3 threshold). Within those rare low upper bound cases, only ~25% stem from preprocessing, whereas ~75% are caused by annotation noise. Thus, preprocessing contributes even less than annotation noise and is not the dominant bottleneck. Its actual impact on overall performance is very small.
> >
> > We greatly appreciate the reviewer’s time and detailed feedback. We hope this resolves the remaining doubts.

---

### Official Review · Reviewer_cMwu · 2025-11-01

**Soundness:** 3
**Presentation:** 3
**Contribution:** 3
**Rating:** 6
**Confidence:** 5

**Summary:**

This work addresses the misalignment between textual descriptions and visual coordinates , which often induces hallucinations in Vision-Language Models (VLMs) for the Spatial-Temporal Video Grounding (STVG) task. The authors propose an "object-centric visual prompting paradigm" : via a pre-processing pipeline of detection , tracking , and ReID , each object is assigned a unique, temporally consistent ID. This reformulates per-frame coordinate prediction into a compact instance-level identification problem. Building on this, the paper introduces STVG-R1 , the first reinforcement learning framework for STVG , which employs a task-driven reward to jointly optimize temporal accuracy, spatial consistency, and structural format regularization . Experiments show the approach achieves SOTA and exhibits strong zero-shot generalization on the unseen multi-object referring video object segmentation task.

**Strengths:**

•	Novelty: Reformulating STVG from dense per-frame coordinate prediction into a "compact instance-level identification task", a novel idea that effectively avoids the difficult problem of VLMs handling coordinate prediction.

•	Novel RL Framework: Proposing STVG-R1, the first reinforcement learning framework for STVG, which employs a task-driven reward to optimize the VLM's reasoning.

•	State-of-the-art Results: Achieves new SOTA performance on multiple STVG benchmarks.

•	Strong Generalization: Exhibits SOTA zero-shot performance on the unseen multi-object referring video object segmentation task (MeViS), highlighting the method's robust generalization ability.

**Weaknesses:**

**Regarding the nature and robustness of the "visual prompting" pipeline:**

•	The pipeline is essentially a complex, training-free data pre-processing pipeline reliant on external SOTA models such as YOLO, SAM2, and ReID, rather than a novel model component.

•	The robustness of this pipeline is not discussed. For example, what happens when detection, tracking, or ReID fail? Many critical details of the pipeline, such as the arbitration logic between components, are missing, which hinders reproducibility.

•	The visual prompts themselves introduce 'visual pollution' and occlude critical information. Embedding ID characters into video frames is a lossy operation that can obscure key details of an object (e.g., facial expressions, specific markings), thereby hindering the model's understanding. Experiments also show the method is sensitive to hyperparameters such as font size, which further confirms the risk of introducing visual interference.

•	The paper does not quantify the additional computational and storage overhead introduced by this complex pipeline. How much (per-video) computational and memory overhead does running YOLO/SAM2/ReID add before using the VLM? This is crucial for assessing the method's practical usability.

**Lack of ablation studies:**

•	The paper provides no ablation study for the three components （r_t, r_s r_f） of the RL reward function R(o), making it impossible to determine the key factors driving the performance improvement.

•	Lack of ablation on the necessity of each component in the pre-processing pipeline.

**Contribution Positioning:**

•	The paper's RL algorithm (GRPO) is heavily borrowed from DeepSeek-R1, which is more of an "application-level innovation" rather than an "algorithmic innovation", and this should be clearly stated in the manuscript.

**Clarity and Justification of Key Mathematical Formulations:**

•	Regarding the spatial reward function r_s(o) in Equation (5): This reward function is designed as a sparse, binary (0 or 1) signal, which prevents the model from receiving "partially correct" feedback (e.g., for predicting a spatially adjacent but incorrect ID). The authors should justify why this sparse reward was chosen over a smoother, continuous reward that could reflect the spatial proximity of the predicted ID to the ground truth, and discuss the considerations for RL training stability and efficiency.

•	Regarding the majority voting rule in Equation (3): This formula determines the global target ID A via majority voting, which is a strong heuristic assumption. This assumption may fail for queries describing transient events (e.g., "a person who flashes by"), potentially leading to incorrect training labels. The paper should discuss the limitations of this design and its potential impact on performance.

•	Regarding the total reward function R(o) in Equation (6): The paper combines temporal and spatial rewards via simple addition, mathematically treating them as independent optimization objectives. However, the evaluation metric for spatio-temporal grounding (vIoU) is inherently coupled. The authors need to explain why a decoupled reward was chosen for training and whether there is a potential misalignment between this design and the final evaluation goal.

**Questions:**

See the questions above

---

> ### Author Response · Authors · 2025-11-21
> **Response To Reviewer cMwu [1/5]**
>
> Thanks for your insightful and thorough review. We will address your concerns one by one.
>
> **Regarding the nature and robustness of the "visual prompting" pipeline**
>
>
> > **W1.** The pipeline is essentially a complex, training-free data pre-processing pipeline reliant on external SOTA models such as YOLO, SAM2, and ReID, rather than a novel model component.
>
> **R1.** The objective of this work is not to propose a new detection or segmentation architecture, but to introduce a new paradigm for spatio-temporal video grounding (STVG). To our knowledge, this is the first work that incorporates visual prompt into STVG, transforming the task from dense coordinates regression into a discrete instance-level ID reasoning problem that general VLMs naturally handle well. This reformulation enables general VLMs to perform STVG without any task-specific visual heads or decoder blocks. While future work may explore integrating the models in the "visual prompting" pipeline into a single unified model, such refinements are beyond the scope of the conceptual contribution we aim to highlight in this paper.
>
> > **W2.** The robustness of this pipeline is not discussed. For example, what happens when detection, tracking, or ReID fail? Many critical details of the pipeline, such as the arbitration logic between components, are missing, which hinders reproducibility.
>
> **R2.** We thank the reviewer for the valuable comment. We analyzed the robustness of the "visual prompting" pipeline under detection, tracking, and ReID failed cases.
>
> **Detection failure can be global or local.** If an object is missed in all frames in the video, no ID is assigned and the model is unable to predict it. Our manual analysis of samples with low upper bound scores shows that cases where the ground-truth target receives no assigned ID account for fewer than 1% of all samples. When detections are missing only in isolated frames, the periodic re-detection mechanism together with bidirectional SAM2 tracking usually recovers them, so the impact remains minimal.
>
> **Tracking or ReID failure** essentially assign different IDs to the same object across frames. However, their influence is incremental rather than catastrophic. **During training,** supervision is derived frame-by-frame and then aggregated across the video by majority voting, which makes the final label robust to occasional frame-level mismatches. **During evaluation,** when the predicted ID is missing in some frames within the model-predicted temporal interval, we apply an ID-repair mechanism that propagates the nearest valid ID using IoU-based matching, further mitigating the effect of local tracking or ReID errors. The ID-repair mechanism is detailed in Appendix A.5 Algorithm 1 in the updated manuscript.
>
> The details of the "visual prompting" pipeline are as follows. We use YOLOv12-x as the detection model with a confidence threshold of 0.25 and SAM2.1-large as the tracking model. For each video, the first frame is passed through the detection model to obtain initial bounding boxes, which are then used to prompt SAM2 for high-quality masks. These masks are propagated across the whole video using the tracking capability of SAM2. To discover newly appearing objects, we run the detection model every 15 frames and compare the detected instances $b_{\text{det}}$ on that frame with the objects tracked to the same frame $b_{\text{trk}}$. A detection is treated as a new instance only if it has low geometric overlap with all tracked objects, measured by the following two criteria:
> \begin{align}
> \operatorname{IoU}(b_{\text{det}}, b_{\text{trk}}) < 0.4,\qquad
> \frac{\operatorname{Area}(b_{\text{det}} \cap b_{\text{trk}})}{\min\left(\operatorname{Area}(b_{\text{det}}), \operatorname{Area}(b_{\text{trk}})\right)} < 0.6.
> \end{align}
>
> If a new instance is detected, we run SAM2’s forward and backward tracking from that frame to recover its full temporal trajectory and integrate the resulting masks into the global mask database. We have updated the manuscript to include arbitration rules and details in both the Method and Experiments sections to ensure that the entire pipeline is fully reproducible.

---

> > ### Author Response · Authors · 2025-11-21
> > **Response To Reviewer cMwu [2/5]**
> >
> > > **W3.** The visual prompts themselves introduce "visual pollution" and occlude critical information. Embedding ID characters into video frames is a lossy operation that can obscure key details of an object (e.g., facial expressions, specific markings), thereby hindering the model's understanding. Experiments also show the method is sensitive to hyperparameters such as font size, which further confirms the risk of introducing visual interference.
> >
> > **R3.** In practice, commonly used STVG benchmarks (HCSTVG-v1/2 and ST-Align) primarily rely on object category, motion trajectory, and spatial relations, rather than fine-grained details. In addition, **any occlusion typically occurs only in isolated frames, and the probability of key visual cues consistently polluted across the entire video is extremely low,** resulting in negligible impact on instance-level grounding.
> >
> > To quantitatively verify these observations, the ablation study of visual prompt designs in Table 5 in the manuscript indicates that **the temporal metric (m_tIoU) is largely insensitive to font size, varying by less than 1% across font sizes 10–40.** This stability suggests that for tasks not dependent on visual prompt, the visual occlusion introduced by the them have only minimal influence. Furthermore, the results in Table 1 show that adding visual prompts has minimal impact on temporal grounding across datasets and models, and **the effect is not consistently negative.** InterVL3-8B and Qwen3-VL-8B exhibit slight improvements, while Qwen2.5-VL-7B and Qwen2.5-VL-72B show small decreases.
> >
> > > **W4.** The paper does not quantify the additional computational and storage overhead introduced by this complex pipeline. How much (per-video) computational and memory overhead does running YOLO/SAM2/ReID add before using the VLM? This is crucial for assessing the method's practical usability.
> >
> > **R4.** **Storage overhead.** The storage overhead introduced by our pipeline is minimal. After preprocessing, each video produces a small folder of approximately **1.3 MB,** containing per-frame instance IDs and bounding boxes for evaluation.
> >
> > **Computational overhead.** We evaluated runtime on the 20s duration HC-STVG videos using a single RTX 4090, averaging results over 100 samples. Inference with our STVG-R1 model takes about 81.47s per video. The preprocessing "visual prompting" pipeline takes 115.24s at the original FPS, with SAM2 contributing roughly 87% of total time. Reducing the input to 5 FPS decreases preprocessing time to 38.10s while preserving comparable accuracy, as the m_vIoU decreases only marginally from 39.7% to 38.9%. Under this setting, preprocessing becomes approximately half as expensive as the VLM's inference time.

---

> > > ### Author Response · Authors · 2025-11-21
> > > **Response To Reviewer cMwu [3/5]**
> > >
> > > **Lack of ablation studies:**
> > >
> > > > **W5.** The paper provides no ablation study for the three components (r_t, r_s r_f) of the RL reward function R(o), making it impossible to determine the key factors driving the performance improvement.
> > >
> > > **R5.** We thank the reviewer for raising this important point. Our analysis of the reward components is summarized as follows.
> > >
> > > **1. Ablation of the spatial reward r_s**
> > >
> > > We performed the complete ablation for r_s. As shown in Table A, **removing r_s reduces spatial accuracy**. However, the model still retains moderate STVG ability, which highlights that the visual prompt enables a general VLM to perform instance-level reasoning.
> > >
> > > Table A. Effect of removing r_s on HCSTVG-v1 test set (%).
> > >
> > > | Reward | m_tIoU | m_vIoU | vIoU\@0.3 | vIoU\@0.5 |
> > > |-------|--------|--------|-----------|-----------|
> > > | r_t + r_f | 56.7 | 36.1 | 61.3 | 28.5 |
> > > | r_t + r_s + r_f | **56.9** | **39.1** | **66.7** | **38.6** |
> > >
> > > **2. Ablation of the temporal reward r_t**
> > >
> > > We also conducted a preliminary ablation of r_t (trained for 400 steps). **When removing r_t, the m_tIoU drops from 38.7 (zero-shot) to 37.5,** while the spatial metrics remain nearly unchanged. This behavior is expected given the design of the spatial reward:
> > > \begin{align}
> > > r_s(o) = \mathbf{1}\big[\iota = \iota^{*} \land \iota \text{ appears in } [t_s, t_e]\big].
> > > \end{align}
> > >
> > > Without a temporal reward to constrain the predicted temporal segment, the model can increase its spatial reward simply by predicting longer temporal intervals, ensuring the chosen ID appears within that predicted $[t_s,t_e]$.
> > >
> > > **3. Ablation of format reward r_f**
> > >
> > > Surprisingly, removing the r_f results in a training curve that is nearly identical to the full-reward setting, as shown in Appendix A.9 Figure 15. This effect is largely due to the simplicity of our format constraint. The desired output structure can already be reliably induced through suitable text prompt. And the base model (Qwen2.5-VL) already supports the <think> and <answer> tokens. As a result, the additional supervision from **r_f provides only limited benefit.** Full results will be reported in the Appendix once training completes.
> > >
> > > Overall, these ablations indicate that **both the spatial reward r_s and the temporal reward r_t are essential,** as they jointly govern correct instance selection and accurate temporal localization. In contrast, the **r_f plays only a minor role**.
> > >
> > > > **W6.** Lack of ablation on the necessity of each component in the pre-processing pipeline.
> > >
> > > **R6.** Thank you for the helpful suggestion. We added ablation studies to evaluate the necessity of the key components in the pre-processing pipeline. As shown in Table B, **periodic re-detection is crucial,** and removing it leads to a notable drop in all vIoU metrics. This degradation arises from the fact that relying solely on the initial detections is insufficient to capture the later-appearing instances. By contrast, **removing backward tracking results in a smaller degradation.** Once a new instance is detected by periodic re-detection, using only forward tracking is still able to recover most of its trajectory. This suggests that backward tracking can be omitted in scenarios where strict runtime efficiency is required.
> > >
> > > Table B. Ablation study of the visual prompting pipeline components on HCSTVG-v1 test set (\%).
> > >
> > > | Method | m_tIoU | m_vIoU | vIoU\@0.3 | vIoU\@0.5 |
> > > |-------|--------|--------|-----------|-----------|
> > > | w/o re-detection | 56.1 | 27.8 | 45.4 | 17.4 |
> > > | w/o backward tracking | 56.8 | 28.4 | 66.2 | 37.1 |
> > > | Full pipeline | **56.9** | **39.1** | **66.7** | **38.6** |
> > >
> > > **Contribution Positioning:**
> > >
> > > > **W7.** The paper's RL algorithm (GRPO) is heavily borrowed from DeepSeek-R1, which is more of an "application-level innovation" rather than an "algorithmic innovation", and this should be clearly stated in the manuscript.
> > >
> > > **R7.** We agree that our method does not introduce a new RL algorithm. Our contribution lies in applying GRPO to STVG for the first time and in designing STVG-specific rewards. We have updated the manuscript to make this explicit in both the Introduction and Related Work sections as:
> > >
> > > "We propose STVG-R1, the first reinforcement learning framework for spatial–temporal video grounding, built upon the GRPO algorithm."
> > >
> > > "To bridge this gap, STVG-R1 integrates GRPO with STVG-specific rewards, achieving superior performance."

---

> ### Author Response · Authors · 2025-11-21
> **Response To Reviewer cMwu [4/5]**
>
> **Clarity and Justification of Key Mathematical Formulations:**
>
> > **W8.** Regarding the spatial reward function r\_s(o) in Equation (5): This reward function is designed as a sparse, binary (0 or 1) signal, which prevents the model from receiving "partially correct" feedback (e.g., for predicting a spatially adjacent but incorrect ID). The authors should justify why this sparse reward was chosen over a smoother, continuous reward that could reflect the spatial proximity of the predicted ID to the ground truth, and discuss the considerations for RL training stability and efficiency.
>
> **R8.** We thank the reviewer for the insightful question. We did experiment with a continuous meanvIoU  (vIoU averaged over frames) as the spatial reward, but it performed worse than our sparse reward design. Tables below show the comparison on HCSTVG-v2.
>
> Table C: Different steps evaluation with **continuous meanvIoU spatial reward** on HCSTVG-v2 val set (%) with 2048 tokens, 1/3 filtering threshold and 2FPS.
>
> | steps | m_tIoU | m_vIoU | vIoU\@0.3 | vIoU\@0.5 |
> |-------|--------|--------|-----------|-----------|
> | 2200  | 59.7   | 38.3   | 64.9      | 35.6      |
> | 2600  | 58.6   | 37.8   | 64.1      | 33.0      |
> | 3000  | 61.6   | 39.9   | 67.1      | 37.4      |
> | 3395  | 60.2   | 38.9   | 66.6      | 36.1      |
>
> Table D: Different steps evaluation with **sparse spatial reward** on HCSTVG-v2 val set (%) with 2048 tokens, 1/3 filtering threshold and 2FPS.
>
> | steps | m_tIoU | m_vIoU | vIoU\@0.3 | vIoU\@0.5 |
> |-------|--------|--------|-----------|-----------|
> | 2200  | 59.8   | 38.7   | 65.4      | 35.3      |
> | 2600  | 60.8   | 39.7   | 67.1      | 37.7      |
> | 3000  | 61.5   | 40.1   | 67.9      | 38.5      |
> | 3395  | 62.0   | 40.2   | 67.8      | 38.8      |
>
> **The continuous reward does not provide better performance, and in fact introduces instability during RL training.** We summarize the reasons as follows:
>
> 1. In our paradigm, the binary signal better matches our instance-level formulation, where the objective is to **choose the most correct instance rather than a relatively correct one.** A continuous vIoU reward blurs this objective.
> 2. In our framework, the VLM does not predict bounding boxes or masks. All geometric coordinates information is produced by the offline "YOLO+SAM2" preprocessing pipeline. **A continuous reward introduces noise that are unrelated to the VLM’s decision,** undermining the stability of the RL optimization.
>
> Regarding training efficiency, **both reward designs exhibit similar convergence behavior.** The "spatial reward + tempral reward" rises rapidly during the first 500 steps, and then improving more gradually. Thus, the sparse reward does not reduce sample efficiency while providing more stable optimization.
>
> > **W9.** Regarding the majority voting rule in Equation (3): This formula determines the global target ID A via majority voting, which is a strong heuristic assumption. This assumption may fail for queries describing transient events (e.g., "a person who flashes by"), potentially leading to incorrect training labels. The paper should discuss the limitations of this design and its potential impact on performance.
>
> **R9.** We agree that majority voting is a heuristic mechanism. However, it does not negatively affect transient events. **Importantly, voting is applied only within the ground-truth temporal interval, not across the entire video.** For very short events, this interval contains only a few frames, so the voting outcome remains correct. We have updated Section 3.2 to clarify this design choice and avoid possible misunderstandings as:
>
> "To formally derive A, we establish a frame-level correspondence within the ground-truth temporal interval."

---

> > ### Author Response · Authors · 2025-11-21
> > **Response To Reviewer cMwu [5/5]**
> >
> > > **W10.** Regarding the total reward function R(o) in Equation (6): The paper combines temporal and spatial rewards via simple addition, mathematically treating them as independent optimization objectives. However, the evaluation metric for spatio-temporal grounding (vIoU) is inherently coupled. The authors need to explain why a decoupled reward was chosen for training and whether there is a potential misalignment between this design and the final evaluation goal.
> >
> > **R10.** We thank the reviewer for raising this important point. Although the final evaluation metric (vIoU) couples temporal and spatial quality, we found that **using a coupled reward does not improve performance**. Specifically, we modified our original reward:
> > \begin{align}
> > R(o) = r_{t}(o) + r_{s}(o) + r_{f}(o),
> > \end{align}
> >
> > into a coupled formulation:
> > \begin{align}
> > R(o) = r_{t}(o) + r_{s}(o)r_{t}(o) + r_{f}(o),
> > \end{align}
> >
> > where **the binary spatial reward is multiplied by the temporal IoU** to mimic the structure of vIoU in evaluation.
> >
> > As shown in Table E, the coupled reward leads to a slightly lower performance compared with the decoupled version. The training curves further confirm that both rewards exhibit very similar convergence trends, as shown in Appendix A.9 Figure 16. Our original decoupled reward already supervises spatial and temporal correctness through simple addition. This simple formulation proves sufficient in practice, and the coupled variant does not provide additional benefit.
> >
> > Table E: Comparison of coupled and decoupled reward designs on the HCSTVG-v1 test set (\%).
> >
> > | Reward | m_tIoU | m_vIoU | vIoU\@0.3 | vIoU\@0.5 |
> > |-------|--------|--------|-----------|-----------|
> > | coupled reward | 56.4 | 38.3 | 65.1 | 38.0 |
> > | decoupled reward | **56.9** | **39.1** | **66.7** | **38.6** |

---

### Author Response · Authors · 2025-12-03
**Quick Summary of Contributions and Rebuttal Updates**

Dear Area Chair,

To facilitate your assessment, we summarize below the key strengths highlighted by the reviewers and the major updates during the rebuttal.

The strengths acknowledged across reviews are:

* **Novel STVG Formulation** `(Reviewer cMwu, bVvE, enhd)`: The paper transforms dense per-frame box regression into a compact instance identification task, for the first time enabling general-purpose VLMs to perform STVG competitively.

* **First RL-Based Framework for STVG** `(Reviewer cMwu, enhd)`: STVG-R1 is the first model to apply RL to STVG with simple, stable, and effective discrete spatial and temporal rewards.

* **SOTA Performance** `(Reviewer cMwu, bVvE, enhd)`: The model achieves 62.0% m_tIoU and 40.2% m_vIoU on HCSTVG-v2, matching or surpassing specialized models across several benchmarks.

* **Strong Generalization Ability** `(Reviewer cMwu)`: Despite training only on single-object datasets, STVG-R1 generalizes robustly to multi-object datasets in zero-shot settings.

The key rebuttal updates addressing reviewer concerns are:

* **Influence of Visual Prompts Occlusion** `(Response to Reviewer cMwu, bVvE)`: We evaluated models with or without visual prompts on tasks that do not depend on spatial grounding, confirming that the small centroid-level marker introduces minimal interference. Specifically, (1) on temporal-only grounding tasks (Charades-STA, TVGBench), tIoU shifts remain below 0.5%; and (2) on OCR-centric evaluation (MME-VideoOCR), accuracy differences average 0.5% across 10 OCR tasks.

* **Pipeline Robustness** `(Response to Reviewer cMwu, bVvE)`: A detailed analysis was added on detection, tracking, and ReID reliability. We observe that (1) global detection failures occur in <1% of samples; (2) local missing detections are corrected via periodic re-detection and bidirectional SAM2 tracking; and (3) tracking/ReID mismatches are mitigated through majority voting (training) and ID-repair (evaluation). These results show the pipeline is robust in practice.

* **Reward Ablation** `(Response to Reviewer cMwu, bVvE)`: Targeted ablations verify the necessity and stability of the proposed discrete spatial rewards. We find that (1) removing the spatial reward (r_s) drops m_vIoU from 39.1% → 36.1%; (2) coupling temporal and spatial rewards lowers performance to 38.3% without any stability benefit; and (3) switching to a continuous 3D IoU reward introduces training instability and degrades results to 38.6%. These outcomes support our reward design choices.

* **Additional Benchmarks and Downstream Tasks** `(Response to Reviewer cMwu, bVvE, enhd)`: To demonstrate broader applicability, we expanded experiments and show that (1) the model generalizes well across ego-centric and movie domains; (2) visual prompt benefits video QA and captioning downstream tasks; and (3) STVG-R1 outperforms VLM baselines on VidSTG.

We believe these updates address the reviewers’ concerns and further strengthen the contributions of STVG-R1. We remain committed to fully open-sourcing the framework upon acceptance.

Best regards,

The Authors

---

### Meta-Review · Area_Chair_pVNM · 2026-01-06

**Summary:**

The paper introduces STVG-R1, the first reinforcement learning (RL) framework for Spatial-Temporal Video Grounding (STVG). The core innovation lies in reformulating the traditional dense per-frame bounding box regression task into a more compact instance-level identification task. By leveraging an "object-centric visual prompting" pipeline (Detection, Tracking, and ReID), the model assigns unique IDs to objects, allowing a Vision-Language Model (VLM) to reason over IDs and timestamps. The authors employ the GRPO reinforcement learning algorithm with task-driven rewards to optimize spatial consistency and temporal accuracy. The method achieves state-of-the-art results on HCSTVG benchmarks and demonstrates impressive zero-shot generalization on multi-object tasks like MeViS.

**Reviewer Concerns:**

Initial reviews praised the novelty of the reformulation but raised significant concerns regarding the robustness and practical overhead of the multi-stage pre-processing pipeline. Reviewer cMwu and bVvE questioned whether the visual IDs (text/markers) would occlude critical information (e.g., facial expressions or OCR text) and highlighted the lack of ablation studies for the specific reward functions ($r_s, r_t, r_f$). Furthermore, reviewers noted that the reliance on external models (YOLO, SAM2) might make comparisons with end-to-end baselines unfair and expressed concerns about the stability of RL training with discrete spatial rewards compared to continuous 3D-IoU signals.

**Reviewer Scores:**

The paper received initial scores of 6, 4, and 4. In the rebuttal, the authors provided a high-quality response that systematically addressed these issues. Specifically, they provided new OCR-centric and temporal-only evaluation results showing that the centroid-level markers introduce negligible interference (<0.5% shift). The authors also provided the requested reward ablations, demonstrating that discrete rewards are actually more stable and effective for RL training than continuous 3D-IoU. Additionally, the robustness analysis of the tracking/ReID pipeline and the expanded results on the VidSTG dataset successfully mitigated concerns about the "training-free" components' reliability. Given that the authors have clarified the "application-level innovation" of GRPO and provided solid empirical evidence for their design choices, the Area Chair finds the contribution significant enough for the community and recommends Acceptance.

---

### Decision · Program_Chairs · 2026-01-26

Accept (Poster)